# Asymmetric coding of reward prediction errors in human insula and dorsomedial prefrontal cortex

Colin W. Hoy [1,2,9] ✉, David R. Quiroga-Martinez[2,3,9], Eduardo Sandoval [2], David King-Stephens[4,5], Kenneth D. Laxer[4], Peter Weber [4], Jack J. Lin[6,7] & Robert T. Knight [2,8]

The signed value and unsigned salience of reward prediction errors (RPEs) are critical to understanding reinforcement learning (RL) and cognitive control. Dorsomedial prefrontal cortex (dMPFC) and insula (INS) are key regions for integrating reward and surprise information, but conflicting evidence for both signed and unsigned activity has led to multiple proposals for the nature of RPE representations in these brain areas. Recently developed RL models allow neurons to respond differently to positive and negative RPEs. Here, we use intracranially recorded high frequency activity (HFA) to test whether this flexible asymmetric coding strategy captures RPE coding diversity in human INS and dMPFC. At the region level, we found a bias towards positive RPEs in both areas which paralleled behavioral adaptation. At the local level, we found spatially interleaved neural populations responding to unsigned RPE salience and valence-specific positive and negative RPEs. Furthermore, directional connectivity estimates revealed a leading role of INS in communicating positive and unsigned RPEs to dMPFC. These findings support asymmetric coding across distinct but intermingled neural populations as a core principle of RPE processing and inform theories of the role of dMPFC and INS in RL and cognitive control.

Adaptive behavior requires predicting the stimuli or actions associated with valuable outcomes. Surprising violations of these predictions (i.e., reward prediction errors, or RPEs) are used to learn and update such associations[1]. The scalar value of RPEs has both signed valence (better or worse than expected?), which reinforces either approach or avoidance behavior, as well as unsigned salience (absolute magnitude, or total surprise) that drives motivation, arousal, and motor preparation[2]. Dorsomedial prefrontal cortex (dMPFC) and the insula (INS) are two key

brain regions that respond to both RPE valence and salience[3,4]. These areas have strong anatomical and functional connections and together form the salience network (also referred to as cingulo-opercular control network), which is involved in performance monitoring and integrating feedback to adjust cognitive control[5–9]. However, conflicting reports linking dMPFC and INS activity to a diverse range of signed and unsigned RPE signals have fueled multiple theoretical proposals about their role in reward learning and cognitive control[10–15].

[1]Department of Neurology, University of California, San Francisco, San Francisco, CA, USA. [2]Helen Wills Neuroscience Institute, University of California, Berkeley, Berkeley, CA, USA. [3]Center for Music in the Brain, Aarhus University & The Royal Academy of Music, Aarhus, Denmark. [4]Department of Neurology and Neurosurgery, California Pacific Medical Center, San Francisco, CA, USA. [5]Department of Neurology, Yale School of Medicine, New Haven, CT, USA. [6]Department of Neurology, University of California, Davis, Davis, CA, USA. [7]Center for Mind and Brain, University of California, Davis, Davis, CA, USA. [8]Department of Psychology, University of California, Berkeley, Berkeley, CA, USA. [9]These authors contributed equally: Colin W. Hoy, David R. Quiroga-Martinez. ✉e-mail: colin.hoy@ucsf.edu

Theories of salience network function have focused primarily on explaining dMPFC activity and can be classified into three families. In one family, theories posit dMPFC encodes positive and negative RPEs together as a "common currency" value or utility signal to inform action selection[16–19]. A second family suggests dMPFC is specialized for processing negative RPEs to coordinate responses to threats and pain[20–22]. A third family of alternative theories argue that dMPFC primarily responds to various unsigned salience signals, either to adjust cognitive control[23,24], orient towards novel or surprising stimuli[25], or track uncertainty in the environment related to exploration and foraging[26].

One barrier to adjudicating between these different theories is that studies often assume positive and negative RPEs are represented together on a symmetric, linear scale relative to a single mean expected value. This classical reinforcement learning (RL) model is partly inspired by foundational observations of dopaminergic neurons that increase their firing rate to positive RPEs and decrease it to negative RPEs[27,28]. However, recent single unit studies in animals have demonstrated that different subpopulations of midbrain dopaminergic neurons separately code for positive RPEs, negative RPEs, and unsigned RPE salience[29–33]. These reports require alternative RPE coding strategies to account for this unexplained RPE coding diversity and reconcile theoretical debates on dMPFC and salience network function.

One possible coding strategy entails allowing different neurons and populations to respond to negative and positive RPEs with different strengths. This asymmetric coding principle has been observed in rodents and non-human primates[34,35] and has been used to improve the performance of deep RL models[35–37]. However, whether asymmetric coding underlies signed and unsigned RPE processing in the human cortex is unclear.

Another challenge for assessing theories of neural RPE coding is that non-human primate studies indicate populations of single units tracking positive and negative RPEs are intermingled within dMPFC[17,34,38,39]. These populations can represent information with heterogeneous coding schemes using both increases and decreases of activity[10,40], which can confound valence-specific responses. Common analysis strategies in human neuroscience measure the average activity within a region and are not well suited for resolving overlapping circuits with opposite valence coding and/or directionality of activity changes, particularly for data with lower spatial resolution such as scalp electroencephalography (EEG).

Intracranial EEG (iEEG) recordings with high spatiotemporal resolution overcome some of the limitations of non-invasive human methods. A recent human iEEG study reported an anatomical dissociation between positive and negative RPE processing across regions associated with value-based decision making, including a bias for negative RPEs in anterior INS[41]. However, this study did not record from dMPFC and focused on region-level analyses that may obscure the different contributions of overlapping circuits and diverse coding schemes within each region.

A final important challenge in elucidating dMPFC function is to understand the flow of information in the salience network. Traditionally, dMPFC has been regarded as a control hub where information about task performance, conflict, and reward is computed. However, similar representations of signed and unsigned RPE variables are also reported in the relatively less studied INS[13–15,42,43], and recent human neuroimaging and iEEG evidence suggests that the INS may lead information transfer to dMPFC[44–46]. Experimental designs and computational models that dissociate signed (positive and/or negative) and unsigned RPEs are required to elucidate the role of the INS in RPE processing and communication.

Here, we bridge these gaps between species, recording modalities, analysis methods, and RPE coding hypotheses by testing whether the asymmetric coding principle can dissociate signed positive and negative, as well as unsigned, RPE responses in local populations of human dMPFC and INS. We recorded iEEG data from 10 epilepsy patients with combined coverage in dMPFC and INS while they performed an interval timing task that used difficulty to manipulate expected outcomes and provide the critical dissociation of RPE valence and salience[47–49]. Using high-frequency activity (HFA) power as a marker of local population dynamics[50–53], we compared the performance of three different linear mixed models in explaining single-trial dMPFC and INS responses to positive, negative, and neutral feedback during the task. We contrasted an RPE value model with linear RPE estimates as a classical RL predictor; an RPE salience model with absolute RPE magnitude as a surprise-related predictor; and an asymmetric model in which absolute negative and positive RPE magnitude were entered as separate predictors. In the asymmetric model, different regression slopes for positive and negative predictors would indicate asymmetric coding of RPEs.

We found that the asymmetric model explained behavior and RPE signals in dMPFC and INS better than traditional RPE value and salience models. Furthermore, individual electrode sites showed differential responsiveness to positive and negative RPEs, such that spatially intermingled neuronal populations separately encoded positive RPEs, negative RPEs, and unsigned RPE salience. Signed RPE value coding was relatively rare, arguing against theories claiming dMPFC primarily represents RPEs in a symmetric, linear scheme. Finally, directed connectivity measures suggested positive and unsigned RPE information was primarily transmitted from INS to dMPFC, while negative and signed RPEs showed limited connectivity modulations. These results resolve competing theories of dMPFC function by demonstrating that asymmetric coding enables both valence-specific and unsigned RPE salience signals to coexist within overlapping dMPFC and INS circuits, while also suggesting that INS plays a leading role in positive and unsigned RPE processing within the salience network.

## Results

We collected behavioral data from 10 patients while recording from implanted SEEG and ECoG electrodes in dMPFC (primarily midcingulate cortex with some nearby supplementary motor complex and anterior cingulate sites) and INS (Fig. 1a; see Methods for patient demographics, electrode coverage, and behavior). These patients performed an interval timing task that dissociates valenced RPE value and non-valenced RPE magnitude by using task difficulty manipulations to modulate reward expectations (Fig. 1b). Easy and hard trials were presented in separate blocks with self-paced breaks in between to minimize fatigue. Error tolerance was adjusted after each trial by two staircase algorithms to clamp accuracy at $74.4 \pm 6.9\%$ and $19.5 \pm 2.6\%$ (mean $\pm$ SD) in easy and hard blocks, respectively. This design dissociates outcome valence and probability by manipulating whether wins or losses are surprising, allowing separation of valenced and non-valenced RPE features. Four patients performed a version of the task that delivered neutral outcomes with no response time (RT) feedback on 12% of trials as an additional source of surprise (see Methods).

### Behavioral adaptation to feedback and positive RPEs

In order to quantify valenced and non-valenced RPE features, we used computational modeling of individual patient behavior to derive single-trial estimates of expected value, RPE value, and RPE magnitude. For each patient, we used logistic regression to predict binary win/loss outcomes across the entire session using error tolerance (Fig. 1c). This model yields patient-specific win probabilities for a given tolerance, which was linearly scaled to the reward function (1, 0, or −1 for winning, neutral, or losing outcomes) to quantify expected value for every trial. Single-trial RPE values were computed by subtracting the expected value from the outcome value, and RPE magnitudes were defined as the absolute value of RPEs. Notably, different reward expectations across easy and hard conditions shift the RPE valence of neutral outcomes to negative in easy blocks and positive in hard blocks (see model predictions in Fig. 1d).

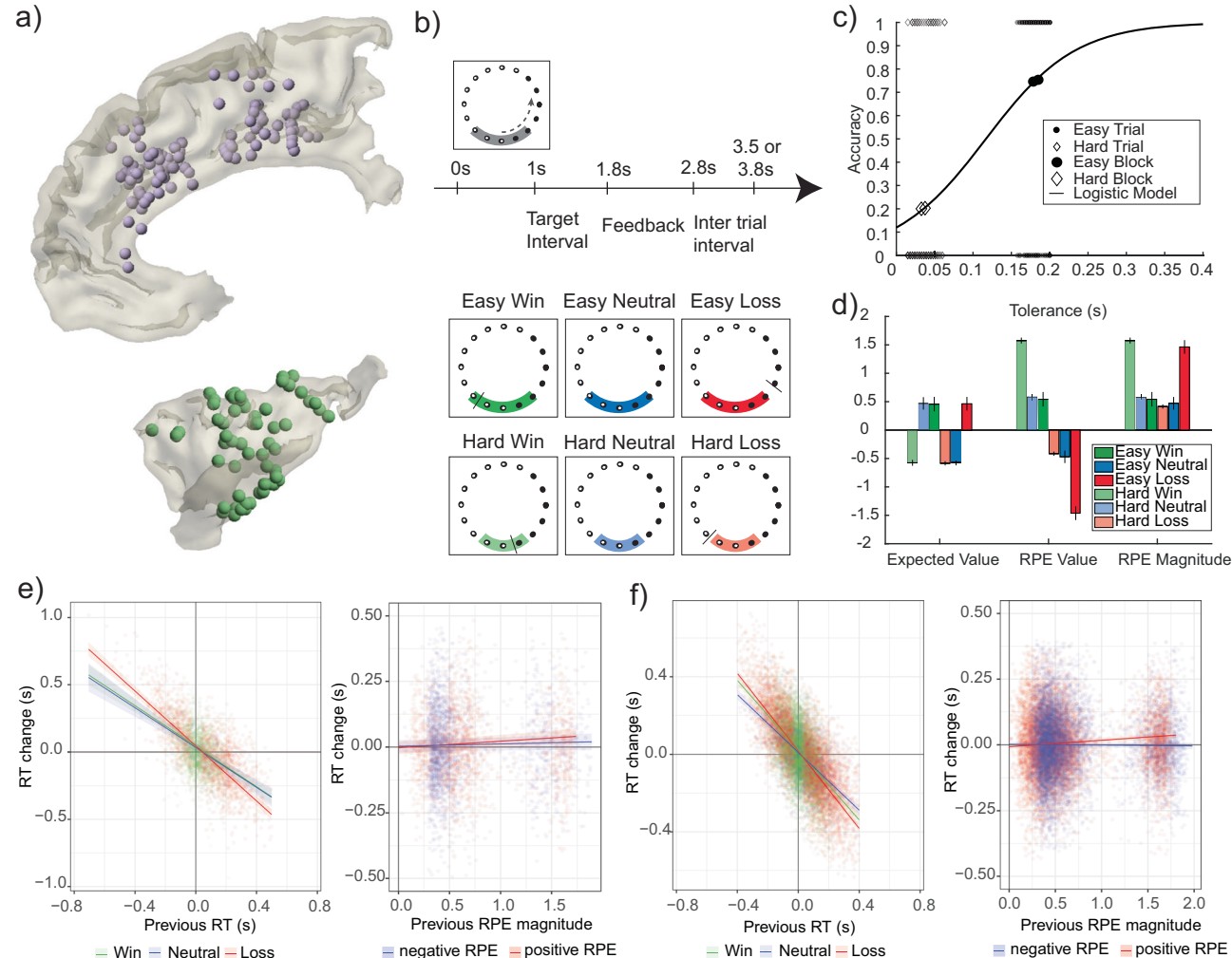

**Fig. 1 | iEEG recording sites, task design, and behavioral modeling.**
**a** Reconstruction of iEEG recording sites in dMPFC (top) and INS (bottom) across all participants plotted on a standardized group brain after mirroring all channels to the right hemisphere for dMPFC and left hemisphere for INS. **b** Participants pressed a button to estimate the time when lights finished moving around a circle. The gray target zone cue displayed error tolerance around the 1 s target interval. Audiovisual feedback is indicated by the tolerance cue turning green for wins and red for losses. A black tick mark displayed RT feedback. For 4 patients, blue neutral feedback was given with no RT marker on 12% of randomly selected trials. **c** Tolerance and outcome data for an example participant. Larger markers show block level accuracy; smaller markers show binary single trial outcomes. Model fit using logistic regression provides single trial estimates of win probability, which is converted to expected value. **d** Predictions for RL model predictors. Error bars indicate group means with standard deviation between participants. Gray dots indicate means for each participant ($n = 10$). **e** Effect of previous RT (left) and previous RPE magnitude (right) on current trial RT change. **f** Convergent results are shown for a larger behavioral dataset from a previous study[49]. Slopes depict regression coefficients. Shaded areas depict 95% confidence intervals around the fitted coefficients. pRPE = positive reward prediction error, nRPE = negative reward prediction error. Source data are provided as a Source Data file.

We investigated the impact that the outcome of the previous trial had on the performance of the current trial. We used linear mixed modeling to predict adjustments in RT relative to the target based on direct RT and outcome feedback, as well as RPEs (Table 1). First, we established that previous trial RTs predicted the adjustment in the current trial ($\chi^2(1) = 2587.269$, $p < 0.001$). Second, we observed a main effect of previous trial outcome (win, neutral, loss; $\chi^2(2) = 12.24$, $p = 0.002$) and an interaction between previous RT and previous outcome ($\chi^2(2) = 43.961$, $p < 0.001$).

Having established relationships between the two sources of feedback and RT adjustments, we next investigated whether RPEs had an impact on behavior. We compared a null model (with only previous RT and feedback predictors, but no RPEs) against an RPE value model (with a signed RPE predictor), an RPE salience model (with an unsigned RPE magnitude predictor), and an asymmetric RPE model (with separate positive and negative RPE magnitude predictors). We found that the asymmetric RPE model predicted RT adjustments better than the null model ($\chi^2(2) = 13.441$, $p = 0.001$) and

the RPE value model ($\chi^2(2) = 10.746$, $p = 0.001$). Compared with the RPE salience model, the asymmetric RPE model fit was not significantly different ($\chi^2(1) = 2.556$, $p = 0.109$), although it performed slightly better according to Akaike Information Criterion (AIC) (Table 1). To corroborate these results and further adjudicate between RPE salience and asymmetric RPE models, we replicated our analyses using RT data from a larger sample of healthy participants ($n = 32$) performing the same task during a previously published EEG experiment (see Methods)[49]. Using the enhanced statistical power in this prior dataset, we found that the asymmetric RPE model predicted RT adjustments better than the RPE salience model ($\chi^2(1) = 58.888$, $p < 0.001$), providing evidence for valence-dependent effects of RPE on behavior.

Coefficients in the asymmetric RPE model of the current iEEG dataset indicated that RT inversely predicted the adjustment in the following trial ($\beta = -0.75$, $p < 0.001$; Fig. 1e; see Supplementary Table 1 for full report of parameters). Thus, if a participant was early, the following RT tended to be longer, whereas if a participant was late, the following RT tended to be shorter, bringing RTs closer to target. An

interaction between previous RT and outcome showed this effect was larger following losses ($\beta = -0.27$, $p = 0.001$), suggesting a win-stay/lose-switch strategy. Lastly, a slowing of RTs was observed after positive ($\beta = 0.02$, $p < 0.001$) but not negative ($\beta = 0.01$, $p = 0.22$) RPEs, supporting a positive bias in the impact of surprising outcomes on behavior. Each of these effects was replicated in the larger sample of healthy participants (Fig. 1f and Supplementary Table 2).

## Positive and negative RPEs are encoded in a separate, valence-specific manner

To determine how neural populations encode RPEs, we assessed how well different sets of RL variables predicted the neural data. We extracted and normalized high frequency band activity (HFA) power from 70-150 Hz at each electrode in dMPFC and INS as a proxy for local population activity (Fig. 2a)[50,53]. Single-trial HFA power was averaged in

50 ms windows sliding by 25 ms from 0 to 600 ms after feedback onset, and these averaged HFA power values were predicted by the different RL variables using linear mixed-effects models across channels and subjects per region and window (Table 2). The resulting fixed-effects model coefficients for each window provide a time series depicting the evolution of the different predictors for a given region.

For both regions, the asymmetric RPE model (separate predictors for positive and negative RPEs) predicted HFA power best, followed by RPE salience (unsigned RPEs) and then RPE value (signed RPEs) (Fig. 2b). Model coefficients (Fig. 2c) indicate RPE value was significantly above zero only in INS ($q_{FDR}$ at peak = .004), while RPE salience was above zero in both regions ($q_{FDR}$ at peak <.001). This suggests HFA power increases with larger RPE magnitudes. However, while the salience RPE model performed well, the asymmetric model described the data best by allowing different coefficients for positive

**Table 1 | Linear mixed-effects modeling of behavioral adaptation**

| Model | Name | Wilkinson notation | Null | Df | $\chi^2$ | p | AIC |
|---|---|---|---|---|---|---|---|
| b0 | Intercept only | RT change ~ 1 + (1|sub) | – | – | – | – | −1217.4 |
| b1 | Previous RT | RT change ~ previous RT + (1 + previous RT|sub) | b0 | 3 | 2587.27 | **<0.001** | −3798.6 |
| b2 | Outcome | RT change ~ previous RT + outcome + (1 + previous RT|sub) | b1 | 2 | 12.24 | **0.002** | −3806.9 |
| b3 | Interaction | RT change ~ previous RT*outcome + (1 + previous RT|sub) | b2 | 2 | 43.96 | **<0.001** | −3846.8 |
| b4 | RPE value | RT change ~ previous RT*outcome + sRPE + (1 + previous RT|sub) | b3 | 1 | 2.69 | 0.1 | −3847.5 |
| b5 | RPE salience | RT change ~ previous RT*outcome + uRPE + (1 + previous RT|sub) | b3 | 1 | 10.88 | **0.001** | −3855.7 |
| b6 | Asymmetric RPE | RT change ~ previous RT*outcome + pRPE + nRPE + (1 + previous RT|sub) | b3 | 2 | 13.441 | **0.001** | −3856.3 |

Model structure is given in Wilkinson notation. Each model was compared with a "null" model excluding the parameter of interest. Degrees of freedom (Df), chi-squared statistics ($\chi^2$) and p-values (p) are reported for one-sided likelihood ratio tests. Models with significantly improved fits have p-values highlighted in bold. Akaike Information Criterion (AIC) is also shown for each model. pRPE positive reward prediction error, nRPE negative reward prediction error, sRPE signed reward prediction error, uRPE unsigned reward prediction error, sub subject. Source data are provided as a Source Data file.

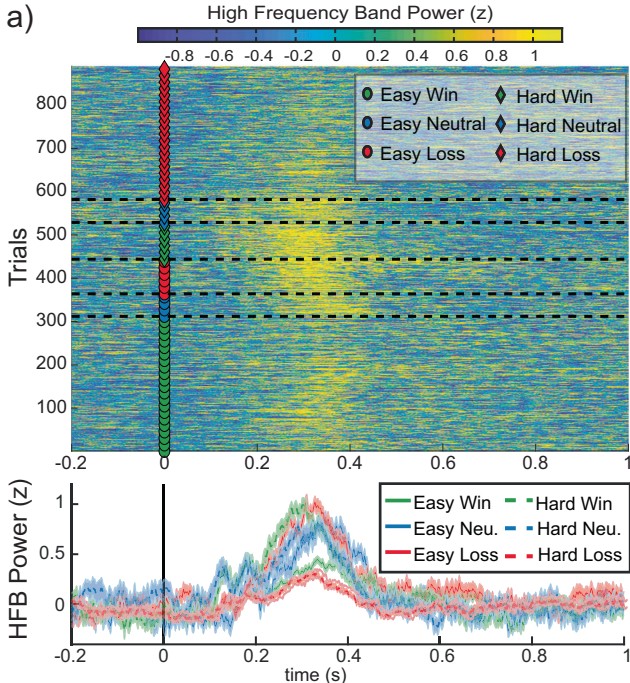

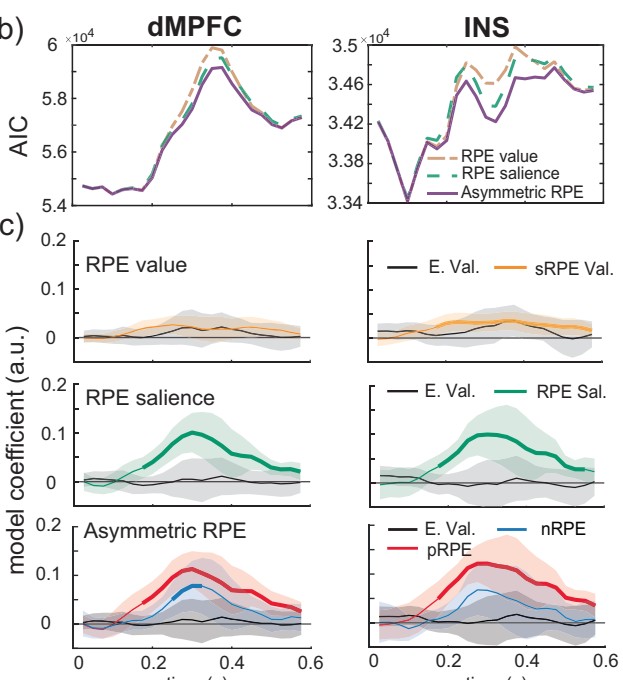

**Fig. 2 | Positive and negative RPEs are encoded in a separate, valence-specific manner. a** Top: Single-trial HFA power at an example channel in the INS is plotted time-locked to feedback (markers at feedback indicate condition). Trials are plotted after sorting by condition, outcome, and RT. Bottom: Condition-averaged HFA power (error bars represent standard error of the mean). **b** Model performance comparison between the three sets of RL variables using Akaike Information Criterion (AIC) for each HFA power time window. Lower values indicate better performance. **c** Region-level, fixed-effects coefficients from linear mixed-effects models predicting single-trial HFA power with three different sets of RL variables: RPE value (expected value + RPE value), RPE salience (expected value + RPE salience), and asymmetric RPE (expected value + positive RPE + negative RPE). HFA power was averaged in 50 ms sliding windows (step size of 25 ms). Significant model coefficients ($q_{FDR} < 0.05$) are plotted in bold. Error bars correspond to 95% confidence intervals. Source data are provided as a Source Data file.

and negative RPEs. Specifically, positive RPEs were associated with an increase in HFA power in both regions, peaking around 275 ms in INS and 300 ms in dMPFC after feedback onset ($q_{FDR}$ at peak <.001). In contrast, the negative RPE effect, although qualitatively similar, was weaker in both regions and significant in dMPFC ($q_{FDR}$ at peak = .036) but not INS ($q_{FDR}$ at peak = .14). The fact that the asymmetric model performs best indicates that RPE value and RPE magnitude alone cannot explain HFA activity and that neuronal populations exhibit asymmetric coding of negative and positive RPEs.

### Diverse responsiveness of neuronal populations to negative and positive RPEs

To understand how different RPE features are coded by neuronal populations in each region, we classified channels with significant responses to positive and/or negative RPEs into four categories (Fig. 3a)[34]. First, we selected positive RPE channels as those significantly

predicted by positive RPE estimates only. Similarly, negative RPE channels were those significantly predicted by negative RPE only. A third category, signed RPE, included channels that responded by significantly increasing their activity with positive RPE, while significantly decreasing activity with negative RPE, or vice versa. Finally, we defined unsigned RPE channels as those that either increased or decreased their activity in response to both positive and negative RPE magnitude. Note that, for each category, RPE can be encoded with both decreases and increases in HFA. This goes beyond classical, bipolar RPE coding in which positive RPEs are represented in activity increases and negative RPEs are represented in decreases (henceforth called "regular coding"). This means that RL theories, including those that allow asymmetric RPE coding (see Discussion for details), account for only three of the eight possible coding strategies using combinations of increases and decreases of HFA (Fig. 3a). Other strategies, such as unsigned RPE coding, as well as cases in which neurons decrease their firing to positive

### Table 2 | Structure of linear mixed-effects model included in the HFA analyses

| Model name | Wilkinson notation |
|---|---|
| RPE value | HFA ~ sRPE + EV + (1 + sRPE + EV \| sub) + (1 + sRPE + EV \| sub:chan) |
| RPE salience | HFA ~ uRPE + EV + (1 + uRPE + EV \| sub) + (1 + uRPE + EV \| sub:chan) |
| Asymmetric RPE | HFA ~ nRPE + pRPE + EV + (1 + nRPE + pRPE + EV \| sub) + (1 + nRPE + pRPE + EV \| sub:chan) |

*EV* expected value, *HFA* high frequency activity, *pRPE* positive reward prediction error, *nRPE* negative reward prediction error, *sRPE* signed reward prediction error, *uRPE* unsigned reward prediction error, *sub* subject, *chan* channel.

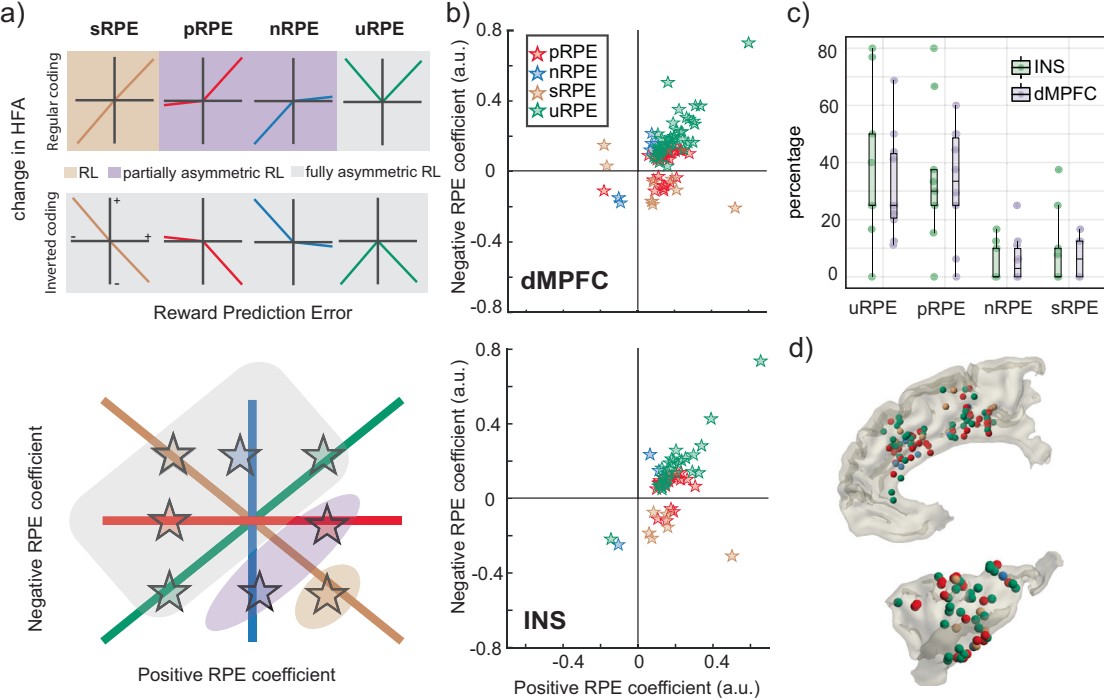

**Fig. 3 | Diverse responsiveness of neuronal populations to negative and positive RPEs. a** Schematic of the different response profiles of local neuronal populations (top), classified into four categories: Positive RPE (pRPE, red), negative RPE (nRPE, blue), unsigned RPE (uRPE, green), and signed RPE (sRPE, gold; see main text for details). For each category, the coding strategy could be regular (i.e., increasing/decreasing HFA power with increasing positive/negative RPE magnitude) or inverted (i.e., decreasing/increasing HFA power with increasing positive/negative RPE magnitude). uRPE populations are labeled by whether they increase or decrease activity regardless of valence. Note that classical RL (gold shade) and current asymmetric RL models (purple shade) only account for regular sRPE, pRPE, and nRPE responsiveness (i.e., partially asymmetric coding). In contrast, unsigned RPEs (i.e., RPE salience) and inverted coding strategies (gray shade) emerge from fully asymmetric coding profiles absent from current theories. Populations

responding to the four categories (colored stars) can be projected on a positive vs. negative RPE plane (bottom). Note how pRPE and nRPE units spread along the x and y axes (positive nRPE coefficients indicate increases in HFA with larger nRPE magnitude), while uRPE and sRPE units spread along the diagonal and off-diagonal. **b** Positive and negative RPE peak coefficients for responsive channels in dMPFC (top) and INS (bottom) belonging to each of the four categories, projected on the two-dimensional RPE plane. **c** Proportion of channels per participant and ROI falling within each category. Box plots depict median and interquartile range across participants (*n* = 10), with whiskers covering most extreme values except outliers. **d** Anatomical location of responsive channels colored per category (Top: dMPFC, bottom: INS). All channels were mirrored to the right hemisphere for dMPFC and left hemisphere for INS. Source data are provided as a Source Data file.

RPE or increase it to negative RPE (henceforth called "inverted coding"), are not accounted for in classical RL. In the following, we evaluate the extent to which different representations arising from asymmetric RPE coding are present in the salience network.

In both regions, the most frequent response profile encoded positive RPE, with a median of 30.00% (IQR = 22.60–44.79) of channels in INS and 33.46%(IQR = 25.00–50.00) of channels in dMPFC (Fig. 3b). A similar proportion of channels encoded unsigned RPE (i.e., RPE salience; MDN = 25.00%, IQR = 22.60–44.79 in INS and MDN = 25.00%, IQR = 20.00–43.75 in dMPFC) while a minority of channels encoded signed RPE (i.e., RPE value; MDN = 0.00%, IQR = 0.00–13.75 in INS and MDN = 6.25%, IQR = 0.00–12.50 in dMPFC) and negative RPE only (MDN = 0.00%, IQR = 0.00–10.63 in INS and MDN = 2.94%, IQR = 0.00–11.1 in dMPFC). When pooling all channels, there were 36 (34%) purely positive RPE channels, 6 (6%) purely negative RPE channels, 9 (8%) signed RPE channels and 39 (37%) unsigned RPE channels among the 106 sites in dMPFC. Similarly, there were 21 (33%) purely positive RPE channels, 3 (15%) purely negative RPE channels, 6 (9%) signed RPE channels and 27 (42%) unsigned RPE channels among the 64 sites in INS. We did not find significant differences in category proportions between regions (all $q_{FDR}$ = .8), suggesting similar coding schemes in INS and dMPFC (Fig. 3c). However, there were differences between categories in the proportion of responsive channels when averaged across regions ($\chi^2(3)$ = 23.86, $p$ < 0.001). Post-hoc, pairwise comparisons revealed higher proportions for unsigned RPE compared to negative RPE ($q_{FDR}$ = .016) and signed RPE ($q_{FDR}$ = .012) and for positive RPE compared to negative RPE ($q_{FDR}$ = .012) and unsigned RPE ($q_{FDR}$ = .012). No other significant differences were found between categories (all $q_{FDR}$ > .8). The proportion of categories did not significantly change along any of the three spatial dimensions (x: $p \geq 0.11$, y: $p \geq 0.24$, z: $p \geq 0.14$), suggesting that they were spatially interleaved. This indicates mixed coding of RPE features across the cortical surface of both regions (Fig. 3d).

Next, we evaluated the extent to which different channels exhibited inverted coding strategies relative to classical RL theories, as defined above. We found that only 1/66 (2%) of unsigned RPE channels decreased activity with both positive and negative RPE magnitude. Similarly, few positive RPE channels (1/57; 2%) and signed RPE channels (2/15; 15%) decreased their activity with increasing positive RPE magnitude. In contrast, 6/9 (66.7%) of negative RPE channels used inverted coding (i.e., increased their activity with increasing negative RPE magnitude). Physiologically, this means that 33.3% of nRPE channels decreased HFA with increasing negative RPE magnitude, as demonstrated by time courses of significant expected value, positive RPE, and negative RPE coefficients for individual channels plotted in Supplementary Fig. 2. This indicates that key variables such as RPE salience and value can be represented by populations of neurons that separately code for negative and positive RPE using both increases and decreases in activity, though coding via decreases in HFA is most prominent for negative RPEs.

### RPE variables predominantly modulate directed connectivity from INS to dMPFC

Given previous reports indicating that INS might lead information transfer in the salience network, we next asked how different RPE variables were communicated between regions by estimating directed functional connectivity between INS and dMPFC. Using cross-correlation, we calculated, for each participant, how well activity in each channel of one region predicted the activity of each channel in the other region, at different time lags. We found that, at the region level, positive and negative RPE magnitude increased correlation between INS and dMPFC channels with a peak lag of 75 ms, indicating that INS activity predicted dMPFC activity best at a 75 ms delay (Fig. 4a).

To investigate communication of RPE variables, we classified between-region channel pairs into the same four categories used for HFA analyses. In this case, channel pairs that significantly decreased or increased their correlation as a function of negative and/or positive RPEs were classified according to their peak coefficient value. We found significant differences between categories in the proportions of channel pairs ($\chi^2(3)$ = 22.10, $p$ < 0.001; Fig. 4b), with a majority responding to unsigned RPE (MDN = 21.88%, IQR = 18.26–34.78) followed closely by purely positive RPE (MDN = 20.00%, IQR = 17.09–25.87). Fewer pairs responded to purely negative RPE (MDN = 1.85%, IQR = 0–6.28) and a minority responded to signed RPE (MDN = 0.00%, IQR = 0.00–3.40). Pairwise contrasts between categories revealed a significantly higher proportion of pRPE compared to nRPE ($q_{FDR}$ = 0.012) and sRPE ($q_{FDR}$ = 0.016); and a significantly higher proportion of uRPE compared to nRPE ($q_{FDR}$ = 0.012) and sRPE ($p$ = 0.039, before FDR correction). This pattern of results is similar to that found in HFA analyses.

To investigate whether the direction of communication was different across RPE features, we next tested for differences in peak lags between RPE categories. We found that lags were predominantly positive, with uRPE (MDN = 100 ms, IQR = 30–180) and sRPE (MDN = 100 ms, IQR = −340–270) having the longest median peak lag for positive RPE coefficients, followed closely by pRPE (MDN = 80 ms, IQR = −50 – 180). For negative RPE coefficients, uRPE had the longest median peak lag (MDN = 100 ms, IQR = 50 – 200 ms) followed by nRPE (MDN = 0 ms, IQR = −260–100) and then sRPE (MDN = −180 ms, IQR = −380–50). This suggests that information predominantly flowed from INS to dMPFC (Fig. 4c). However, we found significant differences in peak lags among categories for both negative ($\chi^2(3)$ = 31.855, $p$ < 0.001) and positive ($\chi^2(3)$ = 9.71, $p$ = 0.02) RPE coefficients. For negative RPE coefficients, sRPE ($q_{FDR}$ < 0.001) and nRPE ($q_{FDR}$ = 0.001) lags were more negative than uRPE lags. For positive coefficients, pRPE lags were slightly more negative than sRPE lags ($q_{FDR}$ = 0.001). These results suggest potential bidirectional communication such that sRPE and nRPE may have also been communicated from dMPFC to INS.

We also observed individual pairs whose correlation showed an inverted coding scheme as defined above. We observed 17/124 (14%) of pRPE pairs and 15/210 (7%) of uRPE pairs decreased their correlation with an increase in their corresponding RPE variable. For nRPE pairs, 22/24 (92%) showed inverted coding relative to classical RL theory, which means only 8% decreased their correlation with increasing negative RPE magnitude. Moreover, 6/11 (55%) of sRPE pairs showed inverted coding by decreasing their correlation with increasing RPE value. Finally, channel pairs involved in RPE communication between INS and dMPFC were also spatially interleaved and category proportions did not change depending on the subregion of dMPFC involved (anterior vs posterior; Supplementary Note 1), which agrees with the aforementioned mixed coding scheme for RPEs in neuronal populations (Fig. 4c).

## Discussion

The valence and salience of RPEs are critical components of reinforcement learning and cognitive control. However, it is unclear how the salience network (dMPFC and INS) represents these variables to facilitate behavioral adaptation. Using HFA power as a proxy for local population activity, we show that a model utilizing asymmetric positive and negative RPE coding explained feedback-related activity in dMPFC and INS better than models including only RPE value or RPE salience. While positive RPE signals were robustly encoded in both regions, negative RPE signals were less prominent and only significant in dMPFC. This positive bias parallels the modulation of RT adaptation by positive but not negative RPEs, which underscores the behavioral relevance of neural RPE coding in these areas. Moreover, neuronal populations at individual channel sites exhibited distinct response profiles, allowing flexible encoding of RPE value and salience with both

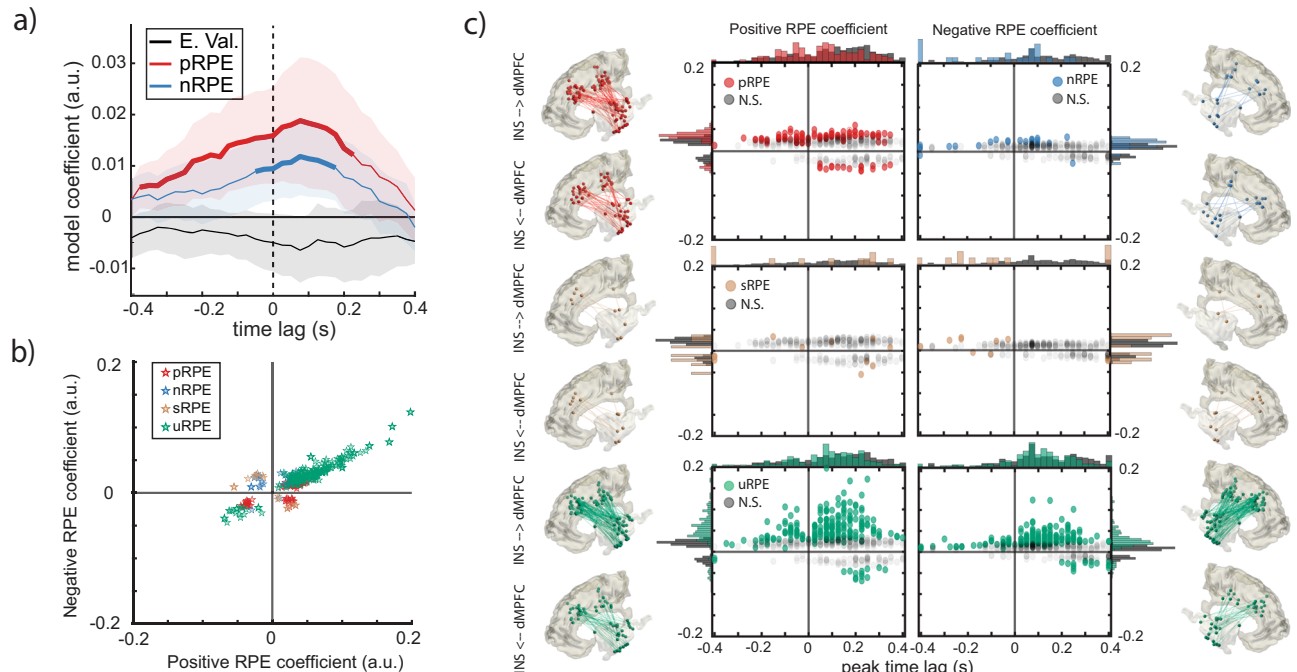

**Fig. 4 | RPE features predominantly modulate directed connectivity from INS to dMPFC. a** Region-level, fixed-effects coefficients depicting the effect of expected value (E. Val.), positive (p) RPE, and negative (n) RPE on the correlation between INS and dMPFC activity at different time lags. Positive lags indicate INS activity precedes dMPFC activity, whereas negative lags indicate dMPFC activity precedes INS activity. Significant model coefficients ($q_{FDR} < 0.05$) are plotted in bold. Error bars correspond to 95% confidence intervals around fixed effect coefficients. **b** Negative RPE and positive RPE peak coefficients grouped by category: Positive RPE (pRPE, red), negative RPE (nRPE, blue), unsigned RPE (uRPE, green)

and signed RPE (sRPE, gold). **c** Peak coefficients and the respective peak time lags, grouped by category. Significantly responsive channel pairs are displayed in color (F test, uncorrected $p < 0.05$). Marginal distributions of peak time lags and coefficients are plotted on x and y axes, respectively. The anatomical location of significant channel pairs is shown for each category, separately for the two directions of communication. Channel positions have been projected to the right/left hemisphere for dMPFC and the left/right hemisphere for INS in the case of positive/ negative RPE coefficients. See Supplementary Fig. 3 for subject-specific connectivity results. Source data are provided as a Source Data file.

increases and decreases in activity. A plurality of channels responded to RPE salience (37% in dMPFC, 42% in INS), as well as purely positive RPEs (34% in dMPFC, 33% in INS). A lower proportion of sites encoded negative RPEs (6% in dMPFC, 15% in INS), and few encoded signed RPE (8% in dMPFC, 9% in INS). This indicates that non-linear, heterogeneous representations of reward information are the dominant coding scheme in dMPFC and INS. Finally, directed connectivity measures indicated channel pairs were primarily modulated by positive RPEs and RPE salience, and that communication for these variables flowed predominantly from INS to dMPFC. Collectively, these results demonstrate that neuronal populations respond differently to positive and negative RPEs, enabling RPE coding diversity in human dMPFC and INS. Below, we discuss how these findings inform theoretical debates in reward learning and cognitive control and their conceptual and methodological implications for principles of neural coding.

Our findings that a model with valence-specific RPEs better explains dMPFC and INS activity has several implications for theories of neural coding and function in the salience network. First, they align with and expand upon recent advances in computational and systems neuroscience by showing that asymmetric coding principles can explain heterogeneous responses to reward and punishment[29,31,34–36]. Asymmetric RPE coding is the strategy used in distributional RL, a recently proposed computational framework that improves model performance by allowing individual units (e.g., neurons) to learn different expected values. This feature takes advantage of neuron-specific learning rates for positive and negative feedback (i.e., asymmetric coding) and enables the population to encode the full distribution of rewards rather than a single mean. Distributional RL has been applied to explain the diverse response profiles of single units in subcortical dopaminergic circuits of rodents[35,36]. Our findings provide

evidence that neuronal populations in the human cortex exhibit asymmetric coding, one of the core principles underlying distributional RL. Future studies using single unit recordings are needed to directly test whether diverse coding schemes for expected value and RPEs in these regions correspond with distributional RL predictions.

Our analyses also revealed that reward and salience information were predominantly represented with increases of HFA power and connectivity. This is consistent with prior studies that observed single units in rodents and non-human primates with elevated firing rates for positive, negative, and salience RPEs in dMPFC[10,11,38,54] and dopaminergic midbrain regions[27,29–33,55]. However, our observation of increasing HFA and connectivity for larger negative RPEs, which accounts for the majority of both RPE salience and valence-specific negative RPE coding, is not accounted for by current RL models (including distributional RL, Fig. 3a), which operationalize negative RPEs as decreases in neuronal activity[35]. Therefore, incorporating asymmetric and inverted coding principles into more biologically consistent RL mechanisms provides an opportunity to enhance representations of reward and salience information in these models.

Notably, we also found representations of larger negative RPEs via decreases in HFA and connectivity, which aligns with classical observations of decreased firing rates of dopaminergic neurons following negative RPEs[27,28]. Importantly, many studies do not typically use asymmetric models that can distinguish this form of bidirectional coding of a single variable using increases and decreases in activity from opponent coding schemes that increase activity to positive versus negative RPEs, which is necessary to avoid confounding interpretations of signed RPE value[48]. Furthermore, spiking and HFA contain complementary but dissociable information[52,53], and they are also modulated by low frequency oscillations[12,56], which may have

different information coding and transmission properties. Future studies examining simultaneously recorded single units, HFA, and LFPs are needed to understand how diverse representations of valence-specific and salience RPEs in individual neurons give rise to asymmetric and inverted coding at the population level.

Importantly, we leveraged the flexibility allowed by asymmetric coding to categorize combinations of positive and negative coefficients corresponding to the key variables underlying central theories of dMPFC function, including RPE value and salience. We found that a plurality of individual sites within and connectivity pairs between dMPFC and INS responded to the salience of RPEs, including many of the strongest responses. This observation supports the proposed central role of dMPFC and INS network in coding salience to enable adjustments of cognitive control[5–7,9,23,25,57]. In contrast to signed RPE theories of dMPFC function[16–19], valenced RPE information was rarely encoded in either neural activity or connectivity responses in a manner consistent with signed RPE value. Notably, our findings are compatible with accounts suggesting dMPFC integrates the positive, negative, and salience RPE variables required to update cognitive control, since separate representations of this information can be read out by downstream regions to support adjustments in approach, avoidance, or motivation. However, our data suggest that most neural populations in dMPFC and INS do not represent these variables together as a combined "common currency" value signal with symmetric but opposite coding of positive and negative RPEs. This finding is consistent with recent proposals that neural representations of value are better understood as related to attention, action plans, goals, vigor, or other choice-related variables[58–60].

Another important result from our channel-level analyses is that diverse populations coding for different RPE variables are spatially interleaved within each region. This observation revealed a more nuanced picture than region-level analyses, which can obscure local heterogeneity within regions. Reconciling population- and region-level results may explain seemingly contradictory evidence supporting different theories of dMPFC function, particularly between single unit studies in systems neuroscience and experiments using functional magnetic resonance imaging or EEG in cognitive neuroscience, which typically average over intermingled populations. Indeed, our population-level results align with non-human primate studies that identified single units sensitive to valence-specific and unsigned salience RPEs within dMPFC[17,34,38], suggesting these circuits are anatomically nonseparable[39]. In contrast, a recent human iEEG study reported an anatomical dissociation between positive and negative RPE processing[41]. However, this study reported region-level effects, leaving the diversity in local population coding of RPE salience and valence unexplored. This emphasizes the need to disentangle spatially intermingled circuits performing different computations within a given region, which is characteristic of previous human iEEG findings in language and attention[61,62]. Furthermore, we found that the proportions of channel sites coding RPE salience, RPE value, and positive and negative RPEs were equivalent in dMPFC and INS. This result supports the view that neural computations underlying reward learning, value-based decision making, and cognitive control unfold in parallel across distributed circuits[58,63–65].

Our results also indicate that population activity within and connectivity between dMPFC and INS have stronger representations for positive than negative RPEs in our task. Modeling RT adaptation showed a slowing effect specifically following positive RPEs that occurred above behavioral adjustments explained by the previous RT and outcome. A recent behavioral study showed effort can enhance learning from positive RPEs and suppress learning from negative RPEs[66], suggesting this neural bias towards positive RPE coding in HFA may reflect the behavioral relevance of those learning signals, particularly in the hard condition. This finding fits with evidence from non-human primate single unit studies[10,17,34] and some human iEEG

results[67]. However, a variety of conflicting evidence from other prior studies argues dMPFC and INS show a bias for processing negative valence[22,41,43,68]. In particular, Gueguen et al. report a bias for negative RPEs in HFA responses in anterior INS[41]. This discrepancy could be due to methodological differences such as region- rather than population-level measures and models that did not control for salience. For example, many of the individual and pairs of channels that responded to negative RPEs in our analyses were revealed to code for salience once we accounted for their response to positive RPEs. Additionally, RPE representations are likely influenced by features of our timing task, including interactions between effort and reward driven by different control demands across easy/hard conditions[66], the absence of learning effects precluding use of traditional temporal difference RL algorithms to estimate value, differences between positive versus negative punishment (i.e., delivering aversive stimuli versus omitting positive rewards), or potential effects of attention and fatigue. Future studies are needed to understand how task demands are integrated with actions to influence the specific relationships between neural activity and RL variables.

Another potential factor influencing the proportion of positive, negative, and salience responses is where our specific recording sites are located relative to functional gradients within dMPFC and INS. For example, the strong representations of salience in our results may be due to the majority of our recording sites falling in mid-cingulate and insular cortices overlapping with the salience network, which is associated with control and performance monitoring[9,57,69,70]. In contrast, single units recorded in non-human primates from anterior cingulate cortex, which is anterior to dMPFC, show reduced salience coding and mostly responded to positive and negative RPEs[34]. This difference in the relative strength of signed and unsigned RPE coding is potentially due to the fact that anterior cingulate cortex is connected to limbic circuits linked to learning, comparing, and choosing values rather than action control[26,70–73]. Similarly, our results showed some negative RPE coding in the INS that aligns with previous studies reporting a bias towards negative RPEs in the anterior portion of this region[4,41,68,74,75]. However, this contrast with the bias towards positive RPE coding found in our INS data may be explained by differences in spatial sampling, which was determined solely based on clinical needs of the patient and covered primarily mid- and posterior INS in our dataset. Interestingly, this potential shift in sensitivity from negative to positive bias across the anterior-posterior axis fits with observations from rodent research of a hedonic "hot spot" in the INS where stimulation induces "liking", which is found posterior to a hedonic "cold spot" in more anterior INS[76,77]. Overall, our converging results from both individual channels and between-region connectivity indicate that dMPFC and INS are predominantly modulated by positive RPEs and RPE salience. Further research with denser sampling within these regions may reveal the fine-grained spatial organization of these RPE variables across subregions.

Lastly, the results of our directed connectivity analyses revealed INS-to-dMPFC communication for positive RPEs and RPE salience processing, which provides direct evidence for hypotheses that the INS plays a leading role in the salience network[44,78,79]. Our findings build upon two recent human iEEG studies showing INS-to-dMPFC connectivity for salience[45,46]. However, these studies used tasks that did not dissociate the valence and salience of feedback. Here, we demonstrate that INS-to-dMPFC directed connectivity predominantly conveys both salience and positive RPE information. Thus, in addition to facilitating salience processing between these two control regions, INS-to-dMPFC communication of positive RPEs may reflect integration of affective information from ventral reward systems including the INS into action processing in dorsal control systems including mid-cingulate cortex[80,81]. Unfortunately, too few channel pairs were significantly modulated to draw firm conclusions about the directionality of negative RPE and RPE value communication. Overall, these results

confirm and expand the role for INS as a source of multiple RPE variables processed in dMPFC, emphasizing the need to shift from an excessive focus on dMPFC towards including the INS in empirical research and theory building.

In conclusion, our results demonstrate that incorporating asymmetric coding principles can capture positive, negative, and salience RPE coding in human dMPFC and INS. Moreover, individual channel analysis strategies similar to those used in non-human systems neuroscience revealed that these populations are interleaved in anatomically overlapping circuits within dMPFC and INS. Importantly, we found that accounting for valence-specific RPE coding using both increases and decreases in activity established that few sites or channel pairs were modulated by signed RPE, arguing against hypotheses that these regions integrate reward and punishment into a common value signal. Instead, our results support a combination of valence-specific and unsigned salience theories of dMPFC and INS function. Finally, our directed connectivity results emphasize the leading role of the INS in both positive and unsigned RPE processing. Overall, these findings bridge region-level analyses common in human neuroscience with population-level analyses in animal models and inform theories regarding neural coding of RPEs in dMPFC and INS.

## Methods

### Participants

Data was collected from ten patients undergoing neurosurgical treatment for medically refractory epilepsy (mean ± SD [range]: 35.2 ± 13.1 [21-57] years old; 1 woman; see Table 3 for patient demographics and electrode coverage). Patients were implanted with stereotactic (SEEG) or subdural grid or strip (ECoG) electrodes, and electrode placement and medical decisions were determined solely by the clinical needs of the patient. Patients were observed in the hospital for approximately a week, and those willing to participate performed the behavioral task during breaks in their clinical treatment. Informed consent was obtained according to experimental protocols approved by the University of California, Berkeley, University of California, Irvine, and California Pacific Medical Center Committees on Human Research. Patients had normal IQ (>85) and spoke fluent English.

### Behavioral Task

The interval timing task was written in PsychoPy[82] (v1.85.3) and consisted of four blocks (two easy and two hard) of 75 trials each (see Fig. 1a for task schematic), with an initial instruction cue before each block started indicating the difficulty level. Two patients completed the task twice, and one patient completed the task three times. The order of block difficulty was fixed as either two easy followed by two hard or alternating from easy to hard (Table 4). Following central fixation and a randomly chosen inter-trial interval ranging from 0.2 to 1.2 s (see Table 4), trials began with presentation of a visual motion cue at a constant speed to arrive at a target at the one-second temporal interval. Participants estimated the interval via button press using the space bar on a keyboard or an RTBox (v5/6) response device[83]. In the first version of the task (n = 6), the motion cue was a blue dot moving continuously upwards in a straight line towards a bullseye target (Supplementary Fig. 1), and in a second version (n = 4), the motion cue was individual lights flickering on then off again in a counter-clockwise order starting and ending at the bottom of a ring of dots on which a gray target zone was centered (Fig. 1b). Participants were instructed that "Your goal is to respond at the exact moment when the ball hits the middle of the target." or "…when [the light] completes the circle." for the first and second versions, respectively. The size of the bullseye in the first version and the width of a gray target zone in the second version indicated the tolerance for successful responses. Veridical win/loss feedback was presented from 1.8 s to either 2.6 or 2.8 s (Table 4) and composed of (1) the tolerance cue turning green/red, (2) cash register/descending tones auditory cues, and (3) a black tick mark denoting the response time (RT) on the ring. Participants received ±100 points for wins/losses. Tolerance was bounded at ± 15–200 or 15–400 ms (Table 4), and separate staircase algorithms for easy and hard blocks adjusted tolerance by −3/+12 and −12/+3 ms following wins/losses, respectively. Participants learned the interval in five initial training trials in which visual motion completed the full linear track or circle. For all subsequent trials, dot motion halted after 400 ms to prevent visuo-motor integration, forcing participants to rely on external feedback. Training concluded with 15 easy and 15 hard trials to initialize both staircase algorithms to individual performance levels. Note that our design minimizes surprise related to task transitions due to the blockwise nature of the difficulty manipulation, presentation of an explicit cue for difficulty level ("Easy"/"Hard") before each block started, and participants' learning of reward probabilities during training. For the second task version, main task blocks introduced neutral outcomes on a random 12% of trials that consisted of blue target zone feedback, a novel oddball auditory stimulus, no RT marker, and no score change.

### Behavioral modeling

The relationship between the tolerance around the target interval and expected value was fit to individual participant behavior using logistic regression. Specifically, tolerance was used to predict binary win/loss outcomes across trials using the MATLAB function *glmfit* with a binomial distribution and logit linking function. Trials with neutral outcomes were not used to fit the models as they were delivered randomly and thus not reflective of performance. The probability of winning ($p_{win}$) for each participant was computed as:

$$p_{win} = \frac{1}{1 + e^{-(\beta_0 + \beta_1 t)}} \quad (1)$$

where $\beta_0$ is the intercept and $\beta_1$ is the slope from the logistic regression, and $t$ is the tolerance on a given trial. Expected value was derived by linearly scaling the probability of winning to the reward function ranging from −1 to 1. RPE value was then computed by subtracting expected value from the actual reward value, and RPE magnitude was computed as the absolute value of RPE value. See Fig. 1c for model predictions by condition. Note that our task minimizes learning by providing an explicit tolerance cue (gray target zone) on each trial after the initial expectations are learned during easy and hard training blocks. Consequently, values were estimated using a logistic regression model instead of traditional temporal difference RL algorithms.

### Table 3 | Patient demographics, electrode coverage, and behavior

| SBJ | Sex | Task Version | Button | Number of Electrodes | | Number of Trials | | Accuracy (%) | |
|-----|-----|------|--------|-------|--------|------|------|------|------|
| | | | | dMPFC | Insula | Easy | Hard | Easy | Hard |
| S01 | M | 1.8.7 | Kb | 16 | 0 | 299 | 297 | 81.6 | 23.9 |
| S02 | M | 1.8.2 | Kb | 16 | 13 | 140 | 138 | 67.9 | 18.8 |
| S03 | M | 1.8.7 | Kb | 18 | 3 | 132 | 149 | 75.0 | 20.1 |
| S04 | M | 1.8.7 | Kb | 17 | 6 | 145 | 149 | 62.8 | 21.5 |
| S05 | M | 1.8.7 | Kb | 5 | 5 | 147 | 147 | 70.1 | 19.7 |
| S06 | M | 1.8.8 | Kb | 8 | 6 | 141 | 133 | 77.3 | 21.8 |
| S07 | F | 2.4.5 | RTBox | 9 | 8 | 144 | 143 | 68.3 | 20.0 |
| S08 | M | 2.4.5 | Kb | 8 | 10 | 145 | 143 | 75.6 | 15.9 |
| S09 | M | 2.4.7 | RTBox* | 8 | 8 | 444 | 446 | 83.9 | 15.6 |
| S10 | M | 2.4.8 | RTBox | 1 | 5 | 286 | 282 | 81.2 | 17.4 |

For Button column, "Kb" indicates responses were collected using the space bar on the built-in laptop keyboard, while "RTBox" indicates a USB button box was used. *For IR87, three runs used the RTbox device, while the keyboard was used to capture responses on the fourth run. Source data are provided in the open data repository.

**Table 4 | Behavioral paradigm parameters**

| Task Version | Motion Cue | Inter-Trial Intervals (s) | Block Order | Error Tolerance Limits (s) | Neutral Outcomes |
|---|---|---|---|---|---|
| 1.8.2 | Linear | 0.5, 0.85, 1.2 | EEHH | 0.2, 0.015 | No |
| 1.8.7 | Linear | 0.2, 0.4, 0.7, 1.0 | EEHH | 0.2, 0.015 | No |
| 1.8.8 | Linear | 0.2, 0.4, 0.7, 1.0 | EEHH | 0.2, 0.015 | No |
| 2.4.5 | Circular | 0.2, 0.4, 0.7, 1.0 | EEHH | 0.2, 0.015 | Yes |
| 2.4.7 | Circular | 0.7, 1.0 | EHEH | 0.4, 0.015 | Yes |
| 2.4.8 | Circular | 0.7, 1.0 | EHEH | 0.4, 0.015 | Yes |

For Block Order, E refers to easy blocks and H refers to hard blocks.

To understand participants' behavioral strategies and their relationship to control and reward variables, we used linear mixed models (R 2022.12.0 + 353, lme4 package 1.1.31) to predict adjustments in RT from previous trial outcomes. RT adjustment was calculated as the difference in RT between the current and previous trial. The first trial of each block and trials following missing responses were dropped from the analysis. In all models, we included by-participant random intercepts and random slopes for the effect of previous trial RT.

A hierarchical model comparison approach was used, starting with a by-subject intercept-only model and building up to full models, as shown in Table 1. Predictors were added incrementally and differences in model performance were assessed using likelihood ratio tests and Akaike Information Criterion (AIC). The computation of $p$-values for the individual coefficients of the winning model (Supplementary Tables 1 and 2) is based on conditional Wald tests with Kenward-Roger approximations using the pbkrtest-package in R. We first tested whether previous trial RT, type of feedback (positive, negative, neutral), or their interaction influenced RT adjustments. Next, we tested whether previous trial RPEs influenced RT adjustments. Here we compared three different models: an **RPE value** model including signed RPEs, an **RPE salience** model including unsigned RPEs, and an **asymmetric RPE** model including positive and negative RPEs as separate predictors. Since RPE value and RPE salience models have the same degrees of freedom, AIC values, but not likelihood ratio tests, were used to compare them. Finally, we replicated these analyses in a dataset from a previous EEG experiment in which 32 healthy participants performed the second version (v2.4.8) of this task[49]. We used the enhanced statistical power of this larger dataset to compare performance of the RPE salience and asymmetric RPE models, as well as to replicate the significant main effects and interactions from the current iEEG dataset.

### iEEG data collection, localization, and preprocessing

The data were recorded at either the University of California Irvine Medical Center ($n = 9$), USA or California Pacific Medical Center ($n = 1$), USA. Patients at Irvine were implanted with stereo-EEG (SEEG) electrodes with 5 mm spacing, and the patient at CPMC was implanted with strips of electrocorticography (ECoG) electrodes with 1 cm spacing. At both sites, electrophysiology and analog photodiode event channels were recorded using a 256-channel Nihon Kohden Neurofax EEG-1200 recording system and sampled at 500 ($n = 3$), 1000 ($n = 3$), or 5000 Hz ($n = 4$). For five patients, analog photodiode channels and a subset of iEEG channels were recorded in a separate Neuralynx ATLAS recording system at Irvine at 4000 ($n = 1$) or 8000 Hz ($n = 4$). For these cases, photodiode events were then aligned to the iEEG data acquired in parallel via the Nihon Kohden clinical amplifier via cross-correlation of shared iEEG channels.

Pre-operative T1 MRI and post-implantation CT scans were collected as part of standard clinical care, and recording sites were reconstructed in native patient space by aligning these scans via rigid-body co-registration according to the procedure described in Stolk et al.[84]. Anatomical locations of electrodes were determined by manual inspection in native patient space under supervision of a neurologist. Electrode positions were then normalized to group space by warping the patient MRI to a standard MNI 152 template brain using volume-based registration in SPM 12 as implemented in Fieldtrip[84]. Group-level electrode positions are plotted in MNI coordinates relative to the cortical surface of the fsaverage brain template from FreeSurfer[85].

Data cleaning, preprocessing, and analyses were conducted using the Fieldtrip toolbox[86] (version d073bb2de) and custom Python (2.7) and MATLAB (2017b) code. Raw iEEG traces were manually inspected by a neurologist for epileptic spiking and spread, as well as artifacts (e.g., machine noise, signal drift, amplifier saturation, etc.). Data in regions or epochs with epileptiform or artifactual activity were excluded from further analyses. Preprocessing included resampling data to 1000 Hz (for datasets recorded at sampling frequencies > 1000 Hz), bandpass filtered using a Butterworth filter from 0.5-300 Hz, re-referenced (bipolar to adjacent electrodes for SEEG data; common average reference across all channels for ECoG data), and bandstop filtered at 60, 120, 180, 240, and 300 Hz (Butterworth filter with 2 Hz bandwidth) to remove line noise and harmonics. Continuous data were then visually inspected to ensure all epochs with artifacts or spread from epileptic activity were removed. Finally, trials were rejected for task interruptions and behavioral outliers (RTs missing, <0.5 s, > 1.5 s, or >3 standard deviations from that patient's mean), resulting in 274-890 trials per patient (mean ± S.D.: 405.0 ± 210.6).

### High frequency broadband power extraction and modeling

Time series data were filtered to high-frequency band activity (HFA) ranges known to correlate with local multi-unit activity[50,52,53]. Specifically, data were segmented from −0.25 to 1.2 s relative to feedback onset, and multitaper time-frequency transformations with 50 ms windows were used to extract power from sub-bands ranging from 70 to 150 Hz in 10 Hz steps. These HFA power values were then log transformed to account for their log-normal distribution[87] in preparation for linear modeling. To normalize these power values against baseline activity, permutation distributions were created for each channel by taking the mean and standard deviation of baseline power values from −0.25 to −0.05 s relative to stimulus onset from 500 iterations of sampling trials with replacement. Feedback-locked power values were then z-scored using the average mean and standard deviation values from those permutation distributions of pre-stimulus baseline power values. This process avoids normalizing HFA power to pre-feedback data which may contain post-response activity and is robust to noisy outlier trials that can skew the baseline data. Finally, sub-bands were averaged together to create a single HFA power time series.

A sliding window approach was then used to average normalized single-trial HFA power values in 50 ms windows stepping by 25 ms from 0 to 0.6 s post-feedback. Mixed-effects models with subject and channel as nested random effects were then used to predict single-trial HFA power data for each time window and brain region. We compared three different RL models using AIC as a performance metric. Note that, due to the large amount of data points, likelihood ratio tests resulted in significant differences between the models at all time points, even after FDR correction, thereby rendering $p$-values uninformative. We therefore relied on AIC values for model comparisons.

All models contained expected value and a unique set of RPE estimates as predictors (Table 2), identical to those used in behavioral modeling: The RPE value model included valenced RPE magnitude estimates (i.e. signed RPE); the RPE salience model included absolute RPE magnitude estimates (i.e. unsigned RPE); and the asymmetric RPE model included separate predictors for positive and negative RPE magnitude estimates. Note that the asymmetric RPE model is mathematically equivalent to a model in which both RPE value and salience are introduced as predictors. That is, RPE value and salience emerge as a linear combination of positive and negative RPE. The asymmetric model was added to operationalize our hypothesis and improve interpretability. Furthermore, in previous work, we have shown that separating positive and negative RPE magnitude helps to disentangle event-related components that are heavily mixed in scalp EEG data[49].

Confidence intervals and two-sided *p*-values for both fixed (i.e. region level) and random (i.e., subject/channel-specific) effects coefficients were obtained from the standard error estimates for each time window. *p*-values of region-level fixed effects were corrected for multiple comparisons across time using the false discovery rate (FDR) methods of Benjamini & Hochberg[88] for each channel. Corrected *p*-values are referred to as $q_{FDR}$ throughout the manuscript. *p*-values of channel-specific random effects were left uncorrected, since the regularizing properties of mixed-effects models result in conservative coefficient estimates that protect against false positives and overfitting. Channels were considered to be significantly predicted by a model regressor if any HFA power window had a model coefficient with $p < 0.05$.

We evaluated the conformity of linear mixed-effects models with the assumption of gaussian residuals using quantile-quantile plots and histograms. We observed a slight skewness of residuals towards the positive end. To ensure correct estimates and inferences, we fitted the models with a robust procedure in which data points with residuals strongly deviating from normality were given less weight during model fitting. In addition, we performed a sensitivity analysis in which HFA values were transformed into "rankit" estimates to ensure normality (see Supplementary Note 2 and Supplementary Figs. 4 and 5 for details). The results converged with the original analysis.

### Estimation and inference on channel responsiveness categories

We classified the channels into four categories according to their responsiveness: (1) positive RPE (increasing/decreasing HFA power with positive RPE magnitude and no significant response to negative RPEs); (2) negative RPE (increasing/decreasing HFA power with negative RPE magnitude and no significant response to positive RPEs); (3) signed RPE (increasing HFA with positive RPE and decreasing HFA power with negative RPE, or vice versa); and 4) unsigned RPE (increasing/decreasing HFA power with both positive and negative RPE magnitude). Because responsiveness changed over time, in a handful of cases a channel could be classified in both the signed and unsigned RPE categories. In those instances, we classified the channel according to the sign of their peak significant coefficients.

To evaluate differences between regions and channel categories, we calculated the proportions of all channels belonging to each category for each subject and region. One subject was excluded from this analysis as they had no electrodes in INS. At the group level, we used Wilcoxon signed-rank tests to compare channel proportions between regions for each category separately. The resulting *p*-values were FDR-corrected across the four between-region tests. After confirming no significant differences between regions for any category, we averaged proportions across regions and tested for differences between categories with a Kruskal-Wallis test followed by post-hoc, FDR-corrected pairwise comparisons with Wilcoxon signed-rank tests. Finally, we used multinomial logistic regression to test for possible spatial gradients in the proportion of categories, employing the x, y and z coordinates of the electrodes as regressors. The multinomial

coefficients of pRPE, nRPE and sRPE were estimated with respect to the uRPE category as a reference.

### Estimation of directed connectivity between INS and dMPFC

We estimated the directed functional connectivity between dMPFC and INS using time-lagged cross-correlation of HFA power time series between all channels in one region and all channels in the other region for each subject. Lags ranged from −400 to 400 ms in 25 ms steps. In our case, positive lags indicate activity in INS precedes activity in dMPFC, whereas negative lags indicate dMPFC activity precedes INS activity. Zero lag indicates no delay between regions.

The resulting correlation-coefficient time-lag series were then predicted by the asymmetric RPE model including expected value, positive RPE magnitude, and negative RPE magnitude as regressors (Table 4). For each time lag, a mixed-effects model was estimated including subject and channel pair as nested random effects. *p*-values for (region level) fixed effects and (channel-pair level) random effects were obtained based on standard error estimates. For fixed effects, *p*-values were FDR-corrected across time-lags for each predictor separately. For random effects, *p*-values were left uncorrected due to the regularizing properties of mixed-effects models. Because the residuals of the model were heavy-tailed, we used robust estimation and sensitivity analyses, as indicated above, to ensure inferences were correct (see Supplementary Note 2 and Supplementary Fig. 6 for details).

For each channel pair and region, we extracted the time lags at which positive and negative RPE magnitude best predicted directed connectivity by finding the peak of the absolute correlation coefficients. We classified channel pairs into the same four categories (pRPE, nRPE, sRPE, uRPE) used for HFA analyses, according to their modulation by negative RPE and/or positive RPE, as indicated above. To evaluate differences in category proportions, we calculated the percentage of all channel pairs belonging to each category for each subject. We tested for differences between categories with a Kruskal-Wallis test followed by post-hoc, FDR-corrected pairwise comparisons with Wilcoxon rank-sum tests. The same statistical procedure was followed to test for differences in peak lags between categories.

### Citation Diversity

Recent work in several fields of science has identified a bias in citation practices such that papers from women and other minority scholars are under-cited relative to the number of such papers in the field[89–92]. Here we sought to proactively consider choosing references that reflect the diversity of the field in thought, form of contribution, gender, race, ethnicity, and other factors. First, we obtained the predicted gender of the first and last author of each reference by using databases that store the probability of a first name being carried by a woman[89,93]. By this measure and excluding self-citations to the first and last authors of our current paper, our references contain 6.1% woman(first)/woman(last), 14.31% man/woman, 19.92% woman/man, and 59.67% man/man. This method is limited in that a) names, pronouns, and social media profiles used to construct the databases may not, in every case, be indicative of gender identity and b) it cannot account for intersex, non-binary, or transgender people. Second, we obtained a predicted racial/ethnic category of the first and last author of each reference by databases that store the probability of a first and last name being carried by an author of color[94,95]. By this measure (and excluding self-citations), our references contain 12.94% author of color (first)/author of color(last), 12.81% white author/author of color, 16.75% author of color/white author, and 57.49% white author/white author. This method is limited in that a) names and Florida Voter Data to make the predictions may not be indicative of racial/ethnic identity, and b) it cannot account for Indigenous and mixed-race authors, or those who may face differential biases due to the ambiguous racialization or ethnicization of their names. We look forward to future work that could help us to better understand how to support equitable practices in science.

**Reporting summary**

Further information on research design is available in the Nature Portfolio Reporting Summary linked to this article.

## Data availability

The raw intracranial and anatomical datasets generated during the current study are not publicly available to preserve patient anonymity. The preprocessed behavioral and intracranial datasets generated and analyzed during the current study are available as a publicly repository in the Zenodo database (https://doi.org/10.5281/zenodo.10023443).[96] The EEG datasets from healthy participants used for behavioral modeling are available in the Open Science Foundation repository and can be found at https://doi.org/10.17605/OSF. IO/JGXFR.[97] Source data are provided with this paper.

## Code availability

Custom Python, R, and MATLAB code used for preprocessing and analysis is available as a GitHub repository (https://github.com/hoycw/asymmetric_RPE_paper), which includes system requirements and dependencies.[98]

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

## Acknowledgements

We thank the participants for their invaluable efforts and I. Griffith for helping develop the paradigm. This work was supported by NINDS R37NS21135 (R.T.K.), CONTE Center PO MH109429 (R.T.K.), Brain Initiative U19NS107609-03 and U01NS108916 (R.T.K., J.J.L.), NIMH F32MH132174 (C.W.H.), NSF GRFP (C.W.H., E.S.), University of California, Berkeley Chancellor's Fellowship (E.S.), and the Independent Research Fund, Denmark (D.Q.M.).The content is solely the responsibility of the authors and does not necessarily represent the official views of the National Institutes of Health.

## Author contributions

C.W.H. and R.T.K. designed the study; C.W.H. collected data; D.K.S., K.D.L., P.W., and J.J.L. managed patients and surgeries; C.W.H., D.R.Q.M., and E.S. analyzed the data; C.W.H. and D.R.Q.M. drafted the manuscript; all authors reviewed, edited, and approved the manuscript; C.W.H., D.R.Q.M., E.S., J.J.L., and R.T.K. acquired funding; and R.T.K. supervised the research.

## Competing interests

The authors declare no competing interests
