## [Peer Review File · Nature Communications]

Asymmetric coding of reward prediction errors in human insula and dorsomedial prefrontal cortexREVIEWER COMMENTS

Reviewer #1 (Remarks to the Author):

In this report the authors present data from a small number of participants with surgically implanted electrodes (iEEG) in dorsomedial PFC and insula. While there were only N=10 individuals who completed this task, sensor coverage over DMPFC and INS was impressive. Methods, analysis, and interpretations were all very well executed. The authors focus on a major theoretical question with these high-quality data: what computational process best characterizes feedback-related activity in these neural systems? This is a timely and important question, as I explain below.

In brief, many experimenters expect to observe bipolar encoding of reinforcement prediction errors in neural circuits (more firing to positive PE and less to negative PE in the same system), predicated on the most basic features of RL and the simplest accounting of dopaminergic midbrain neurons. This isn't intrinsically bad, but it has been an unnecessary limitation in the interpretation of experimental findings. The field of EEG/ERP has been particularly affected by this theoretical perspective (which I am personally against, for what it's worth).

Without going into too much detail, even Holroyd's highly influential RL theory of bipolar feedback-locked activities (Holroyd & Coles, 2002: ~4400 citations) was conceptually re-framed by the author a few years later (2008: 600 citations), yet the field remains in tumultuous debate on what aspect of bipolar signaling to interpret, or if bipolar information encoding is simply incorrect (e.g. Cavanagh & Frank, 2014). New interpretations which objectively test this common assumption are still needed to formally test the defining computational functions of cognitive systems. By the nature of being intracranial recordings with great coverage, lots of data, and interpretable formal computational models, these data offer very definitive statements on the nature of the computational functions of feedback-related circuits.

I have only rather minor comments on this report, all are very addressable:

- 1) The authors set up an analytic test of "asymmetric scaling", which is the formal investigation of unique or common coding of valenced PEs. They align this approach with the new theory of distributional RL. Yet I'm not sure how well scaling asymmetry is a major feature of distributional RL. To my understanding, distributional RL suggests different sensitivities and learning rates across dopamine neurons, which together can represent a population code of the summary statistics of the current reward probabilities (let's call this "scaling"). This is different from simple non-binary encoding of reinforcement where some areas respond to reward, other areas respond to punishers, and other areas (shared or unique) respond to simple surprise (let's call this "asymmetry"). It seems to me the

“asymmetry” aspect is a minor - but not definitive - feature of DRL theory for midbrain dopaminergic neurons. The neuron-specific “scaling” of learning rates seems to be the defining feature of that theory. I urge the authors to ensure that this analytic approach at the level of the local field potential is entirely commensurate with DRL.

2) Lines 233-253 (“...which poses a challenge to both classical RL and dRL.”): At the level of the LFP, even high gamma, it is possible that a decline in power with increasing positive RPE could still align with RL and dRL. Fields are not spikes, and frontal cortex is not SN/VTA. Pretty much nothing occurring in the frontal cortex could “challenge” RL or dRL (i.e. robots could still navigate environments); it could only challenge simplistic applications of these theories to complex neural systems.

3) On a related note, I suggest noting somewhere salient that these findings are limited to high-frequency gamma band power, and lower frequency activities may have different patterns of information coding and information transmission.

4) For negative coding schemes (e.g. less high freq power for increasing PE), I wonder if some of this could be explained by a sensitivity to another process that is inversely correlated with PE. Namely, the state-action value would tend to be inversely correlated with the degree of signed surprise, so any sensor that picked up sustained activity reflective of an eligibility trace for action selection value may appear to inversely code RPE, when in fact it is just doing something else important.

Reviewer #2 (Remarks to the Author):

Hoy and colleagues present a study using intracranial human recordings on reward prediction errors in dMPFC and the insular.

First, I want to acknowledge how hard and time consuming these experiments are and congratulate the authors on the effort.

I am somewhat puzzled by the premise of the study and would like to give my criticism in a more general form. Maybe I am entirely missing the point and I apologize in advance if I do. One of the motivating arguments for the experiment is the described lack of consensus among RPE theories in PFC. First, this is a straw man argument. Yes there is a plethora of findings but most of them stem from different approaches / different techniques and the actual electrophysiological literature is far from

confusing as the authors make it seem. The authors even bring arguments of functional imaging averaging over swaths of cortex yet use HFA as a signal which is equally crude.

Next, the dMPFC as defined by the authors (a huge part of cortex encompassing many medial aspects of PFC and beyond as it looks like) are of course no homogeneous structure. Essentially, the reader is led to believe that there is theoretical considerations of RPE representations within a cortical extend that is more diverse that most even rudimentary imaging methods.

The results are underwhelming. The authors talk about spatial interleaving of different RPE types but what the reviewer sees is non specificity. How can I be sure that the same type of response is not found in other prefrontal regions making the entire exercise futile in terms of specificity? Do the authors actually believe these are unique representations?

If the authors actually wanted to test the distributional dopamine value code hypothesis they would test its actual unique prediction: It should be possible to decode the full value distribution.

Right now the manuscript is confusingly motivated and at best can be interpreted as a translation of findings from the animal literature into the human domain. To that end the reviewer would however be more interested in actually seeing distributions over the entire prefrontal cortex as could be obtained with different techniques. There is a mismatch between the motivation of the study and the actual delivery.

Reviewer #3 (Remarks to the Author):

Review Nat Comm 30012023

This paper tested whether and how neural activity in human dorsal medial frontal cortex and anterior insular region encodes reward prediction errors (RPE), and how functional connectivity between the two regions contributes. The study reports analyses of high gamma power (HFA) from sEEG recordings in 10 patients trained in one of 2 versions of a Target time interval timing task. All but one patient had electrode contacts in both brain regions. The task consisted in block of easy or hard trials which were matched for difficulty levels across subjects.

In order to test relationships between HFA and RPE the authors modelled behaviour using a simple logistic regression model to estimate the probability of winning function. Using such estimation of expected reward outcome, they computed various versions of RPE that would refer to different theoretical propositions about signals observed in those brain regions and about how REP is putatively computed by the brain.

In summary the authors conclude that asymmetric positive and negative RPE coding better explained feedback-related activity in dMPFC and INS, suggesting a bias towards positive reward outcomes in these areas, HFA suggest a bias towards positive reward outcomes in these areas, and that non-linear heterogeneous representations of reward information are the dominant coding scheme. Finally, communication for pRPE and uRPE flowed predominantly from INS to dMPFC.

This is a well-done study with precious data obtained in human subjects with dual recordings in 2 regions known to be relevant for adaptive decision making. However, the current manuscript does not clearly demonstrate how novel the results are in face of the existing electrophysiological literature in monkeys and humans which is quite extensive. The most novel results are the one on directional coding. Some major improvement in the data description, including behaviour, might reinforce the points of the authors. See below comments that might help improve the manuscript.

1) The first major comment is about the core assumption that, in this task, subjects do fluctuate their own representations of expected value based on performance and that their brain estimates RPE based on that representation. Although the authors discuss the fact that their task employs an implicit win-stay lose-shift strategy, the behavioural analysis does not show that subjects adapt. There is no behavioural data showing that subjects adjust their performance within blocks to the current difficulty level (i.e. not that they have varying performance but that they react to changes in performance). Maybe analysing the fluctuations of errors in timing could do the trick or at least provide hints on how subjects conceived the task. It would be essential to then make the hypothesis about the use of some RL algorithm by the brain in the task and to test the various models of RPE computations.

2) If I understand well the authors did not use an RL model but a logistic regression incorporating error tolerance to estimate expected values across a session. This means it is not exactly a single trial estimation but a global estimation adjusted for trial difficulty of each trial; it does not consider local variations of performance due to fatigue, lapses of attention, and does not account for local rate of outcomes. It also assumes a learning rate equal to 1. I understand this is a simplified RPE but it might, or not, have important local differences with RL types of RPE. Some local variations of HFA at outcomes can be observed in figure 2. If this is correct then could the authors discuss whether local variations in performance could impact their analyses of HFA?

3) It is a bit hard to get the specifics of the task design maybe because information is spread in the paper. Since there were 2 versions of a task for different patients and with different specifics (cue, feedback, timings) it would have been great to really show the 2 versions and take a bit of writing to detail every task element. The task procedure is hard to follow, e.g. it is quite difficult to understand

how the motion cue is presented. For instance, in v2, was the target moving continuously or only lighting separated dots as seems to be shown on figure 1? Was it the same for v1? The description of the bullseye target is also very limited. In addition, the task design presented in Figure 1 is the one describing version 2 of the task which was not performed by the majority of patients. The description of a trial goes like this: " trials began with presentation of a visual motion cue at a constant speed to arrive at a target at the one second temporal interval. Participants estimated the interval via button press". The verbal instruction was probably not to estimate the interval of 1 second but to estimate when the cue would reach the arrival zone, which might not be exactly the same (i.e. reaching the border of the zone vs. the centre).

4) The author tested (1223) different patterns of response to positive and negative RPE seeking a validation of models of RPE found in the literature. They report that their data revealed a bias in both regions for unsigned prediction error. Indeed, the overall distribution of significant glmm coefficients (which appears to be most relevant feature) shows a clear extension on the axis of unsigned prediction errors (figure 3b). This comforts the fact that signed RPE cases are the fewest. Regarding the additional bias for pRPE, it is likely that losses/wins in easy and hard have different sources and categorical meanings. The bias for pRPE contrasts somewhat with the literature although that same literature has shown that positive or negative biases are due to tasks statistics and task rules. The bias might in fact reveal that subject do not perceive/use losses as much as they integrate positive outcomes.

5) A general bias in coefficients is clear as well as, in the figure, with a shift towards positive values, i.e. larger increased gamma activity with increased RPE. However, the authors report a high proportion of recording sites with decreasing HFA with increasing positive prediction error. It would be interesting to see a few examples of such decreasing cases. It also raises other questions regarding the activity preceding the outcome response, or other activity potentially reflecting expected value. Did the authors find similar proportion of increasing decreasing HFA with increasing expected values? Actually, for some reasons, analyses of expected value-related HFA are presented only for INS-MPFC coordination not for each region individually.

6) A related comment about HFA: It is said that they are known to correlate with local multi-unit activity; This is true to the extent that sEEG represents a compound signal of large numbers of neurons especially when compared to MUA recorded with single contact microelectrodes. The compounds signal supposedly mixes local populations of single units that do express various coding schemes. In monkeys, Matsumoto et al. 2007 and Quilodran et al 2008 for instance, have shown that single unit activity in the medial frontal cortex reflect most often separately positive and negative outcomes. (Note: the authors referred to Monosov 2017 for RPE coding and anatomical heterogeneity in cingulate recording locations, but I am not sure that paper actually reported precisely that). Many other studies have shown separate cortical encoding of positive and negative outcomes and the modulation by expectations. Quilodran and co-authors also showed HFA reflecting the phenomena. However only a small proportion of units in these studies did not differentiated between positive and negative outcomes. This might explain why the current studies find that asymmetric coding explains better the data and at the same time that

unsigned RPE wins the game – if HFA reflect a compound signal. Maybe the authors could discuss their result in light of these past studies and regarding the nature of HFA vs SUA. Also, some studies showed that non-selective responses to outcomes had lower latencies than the discriminant ones, do the authors find similar effects?

7) In figure 4C it would be also relevant to know how many patients contribute to each MPFC and INS sub-region. In addition, most recordings in INS and in the medial or posterior part of the INS, apparently not reaching Anterior Insula, which however has been the main analysed region in past studies regarding outcome and RPE analyses. The authors might want to discuss that.

8) Figure 4c related data led the authors to suggest that there might be bidirectional communication. This is very likely indeed, and the data for pRPE and uRPE, reveal quite a number of cases with negative lags. The analyses do not discriminate between subregions but the data (recording sites) clearly clusters at least in MPFC between aMCC regions and pMCC or PCC regions. Could the authors check whether subregions in MPFC lead to different ratio of positive versus negative lags with INS?

Minor comments.

1) (I312-320) Isn't it expected (statistically) that the proportions of INS/MPFC correlations coefficients for each RPE category were similar to that found in HFA analyses?

2) Maybe figure 4c would be more straightforward if axes were inverted such that time lags were facing the anatomical representations (i.e. positive time lags would be at the levels of INS->MPFC figures).

Reviewer #4 (Remarks to the Author):

Title: asymmetric coding of reward prediction errors in human insula and dorsomedial prefrontal cortex

Summary: Hoy et al. examine asymmetric scaling of RPEs in epilepsy patients, specifically focusing on high gamma activity in the dorsomedial prefrontal cortex and insula while patients performed a psychometrically regulated time estimation task. The authors find that unsigned and positive RPEs are preferentially encoded, over signed and negative RPEs, and that such activity occurs slightly earlier in insula than dmPFC. This is an interesting study, but is impugned by the lack of controls. Additionally, there are some issues with interpretation of the results that should be corrected.

Major critiques:

- It is disingenuous to frame the findings of this study as relevant for distributional RL without performing several controls and additional analyses. As is, this manuscript is about asymmetric RPE scaling, not dRL. It is unsurprising that the results don't cohere with dRL given the task design and measurements. If the authors read the Supplementary Information for the Dabney study, they will notice numerous controls that were performed to ensure that they were observing distributional RL, and not some other facet of asymmetric reward prediction scaling. Moreover, the current study doesn't meet the basic criteria for establishing the presence of distributional RL: 1) diverse RPE scaling, 2) diverse reversal points, and 3) correlation between 1 and 2. Furthermore, distributional RPE coding at the level of neuronal populations (~0.5 million neurons near an sEEG) is much less tractable, harder to understand. It begs the question, why did the authors mention dRL at all, especially given the observed results (relatively little negative PRE scaling, compared to unsigned RPE scaling)? I would recommend only mentioning dRL in the discussion, if at all, with extensive discussion of the caveats involved with the current study: dRL is not defined at the level of neuronal populations, the authors didn't show distributional encoding, the authors didn't estimate reversal points, or show any neural or behavioral evidence for asymmetric learning.

- In Figure 2, the largest high gamma values correlate with the density of task structure changes. One can even see increases in high gamma earlier in the largest blocks. It therefore seems important to control for the potential surprise induced by changes in the structure of the task, especially, given previous evidence that ACC is sensitive to changes in task structure. The authors should also show the time courses of the TD model variables. How did the authors account for an initial learning period that is ubiquitous in TD learning models.

- The interpretation on line 219 indicates that the asymmetric model performs best, but it appears to perform similarly to the unsigned RPE model. Furthermore, the unsigned RPE model is the most prevalent. More justification for the authors interpretation is required.

- Activity in dmPFC has repeatedly been associated with mental effort and cognitive control. An essential control analysis should therefore account for variance introduced by trial difficulty. It is not clear whether this analysis is possible, given the task design of the current study, as trial difficulty highly correlated with outcome surprise. To what extent does the task design promote expression of these particular flavors of RPEs. This concern is amplified by the observed lack of expected value encoding. One might expect that a brain region responsible for the computations implicated in this study would also represent expected value variables. That these brain areas are not representing expected value raises the concern that they may be representing some other type of surprise rather than task related RPE. Maybe shifting the model fits by one trial to account for subsequent trial effects, or examining responses to errors, could speak to how much of the high gamma activity is induced by cognitive control demands?

Minor critiques:

- It seems like the sentence on line 50 should be supported with citations.

- The introduction could be more concise. Removing the irrelevant dRL mentions and focusing more on asymmetric scaling may help with this.
- The models fit the unsigned RPE data better, yet the effects were smallest for unsigned RPEs. The authors should discuss this discrepancy.
- The anatomical claims in the discussion (line 474) are unsupported by data.
- Line 662: did the authors confirm that their data matched the model assumptions after log-normalizing, with a Lilliefors's test for example?
- Line 667: 'feedback-lock' should be 'feedback-locked'
- General: Both linear mixed-effects models are superficially described. It's unclear if both patient and channel variables were nested relative to the signals, or there was a hierarchical nesting of channel and patient variables. This issue could be clarified by specifying the models in either matrix or Wilkinson notation.

Response to reviewers for Hoy, Quiroga-Martinez, et al. “Asymmetric coding of reward prediction errors in human insula and dorsomedial prefrontal cortex” in *Nature Communications*

Reviewer comments are in black and are numbered by reviewer and comment (e.g., the fourth comment by Reviewer #3 is titled R3.4). Author responses are in blue, and modified text from the revised manuscript is in red.

Reviewer #1 (Remarks to the Author):

In this report the authors present data from a small number of participants with surgically implanted electrodes (iEEG) in dorsomedial PFC and insula. While there were only N=10 individuals who completed this task, sensor coverage over DMPFC and INS was impressive. Methods, analysis, and interpretations were all very well executed. The authors focus on a major theoretical question with these high-quality data: what computational process best characterizes feedback-related activity in these neural systems? This is a timely and important question, as I explain below.

In brief, many experimenters expect to observe bipolar encoding of reinforcement prediction errors in neural circuits (more firing to positive PE and less to negative PE in the same system), predicated on the most basic features of RL and the simplest accounting of dopaminergic midbrain neurons. This isn't intrinsically bad, but it has been an unnecessary limitation in the interpretation of experimental findings. The field of EEG/ERP has been particularly affected by this theoretical perspective (which I am personally against, for what it's worth).

Without going into too much detail, even Holroyd's highly influential RL theory of bipolar feedback-locked activities (Holroyd & Coles, 2002: ~4400 citations) was conceptually re-framed by the author a few years later (2008: 600 citations), yet the field remains in tumultuous debate on what aspect of bipolar signaling to interpret, or if bipolar information encoding is simply incorrect (e.g. Cavanagh & Frank, 2014). New interpretations which objectively test this common assumption are still needed to formally test the defining computational functions of cognitive systems. By the nature of being intracranial recordings with great coverage, lots of data, and interpretable formal computational models, these data offer very definitive statements on the nature of the computational functions of feedback-related circuits.

We thank the reviewer for their kind words and for clearly articulating the importance and prevalence of theoretical debates over the nature of PE coding in cognitive neuroscience, particularly the strong influence that simple, bipolar RPE coding has on many experimenters' interpretations of neurophysiological data.

I have only rather minor comments on this report, all are very addressable:

R1.1) The authors set up an analytic test of “asymmetric scaling”, which is the formal investigation of unique or common coding of valenced PEs. They align this approach with the new theory of distributional RL. Yet I’m not sure how well scaling asymmetry is a major feature of distributional RL. To my understanding, distributional RL suggests different sensitivities and learning rates across dopamine neurons, which together can represent a population code of the summary statistics of the current reward probabilities (let’s call this “scaling”). This is different from simple non-binary encoding of reinforcement where some areas respond to reward, other areas respond to punishers, and other areas (shared or unique) respond to simple surprise (let’s call this “asymmetry”). It seems to me the “asymmetry” aspect is a minor - but not definitive - feature of DRL theory for midbrain dopaminergic neurons. The neuron-specific “scaling” of learning rates seems to be the defining feature of that theory. I urge the authors to ensure that this analytic approach at the level of the local field potential is entirely commensurate with DRL.

We thank the reviewer for raising this point, which is similar to comments of the other reviewers below. We agree that asymmetric coding of positive and negative RPE representations is a core principle underlying dRL theory but does not encapsulate all definitive properties and predictions of dRL (see responses to related points R2.3 and R4.1 for details). Accordingly, we have modified the language throughout our manuscript to deemphasize links with dRL, which are now confined to one paragraph in the Discussion. We have refocused the framing and findings of our study on the theoretical and conceptual implications of asymmetric coding in the context of salience network function without directly testing dRL predictions about relationships between RPE scaling and reversal points. We reframed our arguments to make clear that our data use measures of asymmetric coding (rather than asymmetric scaling) to elucidate neurophysiological representations of signed and unsigned RPEs, which are relevant to but do not directly address dRL.

- **Abstract (lines 30-33):** “Recently **developed RL models allow** neurons **to respond differently** to positive and negative RPEs. Here, we use intracranially recorded high frequency activity (HFA) to **test whether** this **flexible asymmetric coding** strategy captures RPE coding diversity in **human INS and dMPFC.**”
- **Introduction:**
 - **Lines 73-77:** “However, recent single unit studies in animals have demonstrated that different subpopulations of midbrain dopaminergic neurons separately code for positive RPEs, negative RPEs, and unsigned RPE salience²⁹⁻³³. **These reports require** alternative **RPE coding strategies** to account for **this** unexplained RPE coding diversity and reconcile theoretical debates on dMPFC and salience network function.”

- Lines 78-83: “One possible coding strategy entails allowing different neurons and populations to respond to negative and positive RPEs with different strengths. This **asymmetric coding** principle has been observed in rodents and non-human primates^{34,35} and has been used to improve the performance of deep RL models³⁵⁻³⁷. However, whether asymmetric coding underlies signed and unsigned RPE processing in the human cortex is unclear.”
- Lines 109-112: “Here, we bridge these gaps between species, recording modalities, analysis methods, and **RPE coding hypotheses** by testing whether the asymmetric coding principle can dissociate signed positive and negative, as well as unsigned, RPE responses in local populations of human dMPFC and INS.”
- Lines 122-123: “In the asymmetric model, different regression slopes for positive and negative predictors would indicate asymmetric coding of RPEs.”
- Lines 130-132: “Moreover, a substantial portion of channel sites increased activity to negative RPE and decreased it to positive RPE. This “inverted” coding scheme invites future work to expand the scope of **current RL models**.”
- Lines 135-137: “These results resolve competing theories of dMPFC function by demonstrating that asymmetric coding enables both valence-specific and unsigned RPE salience signals to coexist within overlapping dMPFC and INS circuits, ...”
- Results:
 - Lines 294-304: “Note that, for each category, RPE can be encoded with both decreases and increases in HFA. **This goes beyond** classical, **bipolar RPE coding in which** positive RPEs are represented in activity increases and negative RPEs are represented in decreases (henceforth called “regular coding”). This means that **RL theories, including those that allow asymmetric RPE coding (see discussion for details)**, account for only three of the eight possible coding strategies **present in our data** (Fig. 3a). Other strategies, such as unsigned RPE coding and cases in which neurons decrease their firing to positive RPE and increase it to negative RPE (henceforth called “inverted coding”) are not incorporated in **RL**. In the following, we evaluate the extent to which different **representations** arising from asymmetric **RPE coding** are present in the salience network.”
 - Figure 3 caption (lines 338-342): “*Note that classical RL (gold shade) and current asymmetric RL models (purple shade) only account for regular sRPE, pRPE, and nRPE responsiveness (i.e., partially asymmetric coding). In contrast, unsigned RPEs (i.e., RPE salience) and inverted*

coding strategies (gray shade) emerge from fully asymmetric coding profiles absent from current theories.“

- Lines 359-362: “This indicates that key variables such as RPE salience and value can be represented by populations of neurons that separately code for negative and positive RPE using both increases and decreases in activity, resulting in inverted coding schemes with respect to **current RL theories.**”
- Discussion:
 - Lines 458-471: “First, they align with and expand upon recent advances in computational and systems neuroscience by showing that asymmetric coding principles can explain heterogeneous responses to reward and punishment^{29,31,34-36}. Asymmetric RPE coding is the strategy used in distributional RL, a recently proposed computational framework that improves model performance by allowing individual units (e.g., neurons) to learn different expected values. This feature takes advantage of neuron-specific learning rates for positive and negative feedback (i.e., asymmetric coding) and enables the population to encode the full distribution of rewards rather than a single mean. Distributional RL has been applied to explain the diverse response profiles of single units in subcortical dopaminergic circuits of rodents^{35,36}. Our findings provide novel evidence that neuronal populations in the human cortex exhibit asymmetric coding, one of the core principles **underlying distributional RL.** Future studies using single unit recordings are needed to directly test whether diverse coding schemes for expected value and RPEs in these regions correspond with distributional RL predictions.”
 - Lines 487-491: “Notably, current RL models (including distributional RL, Fig. 3a) operationalize positive and negative RPEs as increases and decreases in neuronal activity, respectively³⁵. **Therefore, incorporating asymmetric and inverted coding principles into more biologically consistent RL mechanisms provides an opportunity to enhance representations of reward and salience information in these models.**”
 - Lines 492-495: “Importantly, we leveraged the **flexibility** allowed by asymmetric coding to categorize combinations of positive and negative coefficients corresponding to the key variables underlying central theories of dMPFC function, **including RPE value and salience.**”
 - Lines 592-594: “In conclusion, our results demonstrate that incorporating asymmetric coding principles can capture positive, negative, and salience RPE coding in human dMPFC and INS.”

R1.2) Lines 233-253 (“...which poses a challenge to both classical RL and dRL.”): At the level of the LFP, even high gamma, it is possible that a decline in power with increasing

positive RPE could still align with RL and dRL. Fields are not spikes, and frontal cortex is not SN/VTA. Pretty much nothing occurring in the frontal cortex could “challenge” RL or dRL (i.e. robots could still navigate environments); it could only challenge simplistic applications of these theories to complex neural Systems.

We agree with the reviewer that without spiking of dopaminergic neurons in SN/VTA, our data do not directly address the canonical neurophysiological RL and dRL models originally proposed by Schultz in 1997. We thank the reviewer for pointing this out. We have revised the manuscript to carefully describe previous neurophysiological results, the conceptual contribution of our findings, and how incorporating these coding principles into future RL models may improve their information coding:

- Discussion (lines 472-487): “Our analyses also revealed that reward and salience information were represented with both increases and decreases of HFA activity and connectivity. Prior studies have observed single units in rodents and non-human primates with elevated firing rates for positive, negative, and salience RPEs in dMPFC^{10,11,38,54} and dopaminergic midbrain regions^{27,29-33,55}. However, these studies do not typically distinguish these opponent coding schemes for increased activity to positive versus negative RPEs from inverted coding using decreases in activity. This is important to avoid confounding interpretations of signed RPE value⁴⁸. Inverted coding strategies may be more prominent in HFA than single or multiunit data because HFA aggregates both spiking and subthreshold input activity across local populations, making it sensitive to small fluctuations in firing rates and membrane potentials^{52,53}. Furthermore, spiking and HFA are also modulated by low frequency oscillations^{12,56}, which may have different information coding and transmission properties. Future studies examining simultaneously recorded single units, HFA, and LFPs are needed to understand how diverse representations of valence-specific and salience RPEs in individual neurons give rise to asymmetric and inverted coding at the population level.”

R1.3) On a related note, I suggest noting somewhere salient that these findings are limited to high-frequency gamma band power, and lower frequency activities may have different patterns of information coding and information transmission.

We agree with the reviewer that low frequency rhythms are another important modulator of encoding and transmitting information within and between neural populations, and as noted above, we have incorporated this point into the Discussion:

- Discussion (lines 482-487): “Furthermore, spiking and HFA are also modulated by low frequency oscillations^{12,56}, which may have different information coding

and transmission properties. Future studies examining simultaneously recorded single units, HFA, and LFPs are needed to understand how diverse representations of valence-specific and salience RPEs in individual neurons give rise to asymmetric and inverted coding at the population level.”

R1.4) For negative coding schemes (e.g. less high freq power for increasing PE), I wonder if some of this could be explained by a sensitivity to another process that is inversely correlated with PE. Namely, the state-action value would tend to be inversely correlated with the degree of signed surprise, so any sensor that picked up sustained activity reflective of an eligibility trace for action selection value may appear to inversely code RPE, when in fact it is just doing something else important.

We agree with the reviewer that additional experiments using different tasks and behavioral models will be needed to compare the different computational explanations of HFA, including state-action values. However, we note that modeling integrated tolerance-RT interactions to approximate state-action values would be complex in our continuous task, so this specific issue may be better addressed using tasks with discrete state spaces. Additionally, we do observe EV coding using both increases and decreases in HFA, which supports inverted coding being a general principle of HFA (Sup. Fig. 2). Accordingly, we have added language in the Discussion outlining the influence of some factors of our task, as well as a sentence noting the need to examine relationships between neural activity and a wider range of RL variables.

- Discussion (lines 546-553): “Additionally, RPE representations are likely influenced by features of our timing task, including interactions between effort and reward driven by different control demands across easy/hard conditions⁶⁶, the absence of learning effects precluding use of traditional temporal difference RL algorithms to estimate value, differences between positive versus negative punishment (i.e., delivering aversive stimuli versus omitting positive rewards), or potential effects of attention and fatigue. Future studies are needed to understand how task demands are integrated with actions to influence the specific relationships between neural activity and RL variables.”

Reviewer #2 (Remarks to the Author):

Hoy and colleagues present a study using intracranial human recordings on reward prediction errors in dMPFC and the insular. First, I want to acknowledge how hard and time consuming these experiments are and congratulate the authors on the effort.

We thank the reviewer for their acknowledgement of our efforts.

I am somewhat puzzled by the premise of the study and would like to give my criticism in a more general form. Maybe I am entirely missing the point and I apologize in advance if I do.

R2.1: One of the motivating arguments for the experiment is the described lack of consensus among RPE theories in PFC. First, this is a straw man argument. Yes there is a plethora of findings but most of them stem from different approaches / different techniques and the actual electrophysiological literature is far from confusing as the authors make it seem. The authors even bring arguments of functional imaging averaging over swaths of cortex yet use HFA as a signal which is equally crude.

We thank the reviewer for this important point. We acknowledge that quite a few electrophysiological studies in animals, particularly in non-human primates, have characterized different types of RPE coding in PFC. However, the human literature, most of which has used non-invasive methods, is mixed and less conclusive, due to both methodological challenges and the use of confounded RL models (see comments by Reviewer #1). Given that understanding the human brain and cognition is a fundamental goal of neuroscience, one major contribution of our study is to bridge this gap between species and between invasive and non-invasive recordings, using models that carefully disentangle the valence and salience of RPEs in the context of confounding effects of information encoding using both increases and decreases in activity.

Second, further investigations are warranted even within the animal electrophysiology literature on RPE coding in PFC. Some studies do not use tasks or models that appropriately dissociate effects of valence and salience, and many do not account for confounds related to information coding using both increases and decreases in activity. Importantly, asymmetric RPE models elegantly disentangle these confounds. For example, selective decreases to negative RPEs or increases to positive RPEs can fit a linear RPE model and thus be interpreted as signed RPE value, but asymmetric RPE models will correctly identify these responses as valence-specific. Furthermore, unsigned RPE magnitude and increased firing to negative RPEs are not typically integrated into current RL models, so we hope that highlighting these response patterns will inspire biologically realistic theories of RPE processing and RL algorithms. Overall,

our findings emphasize the need for asymmetric RPE models to address existing ambiguities in the well-developed animal electrophysiology literature.

- Discussion (lines 472-487): “Our analyses also revealed that reward and salience information were represented with both increases and decreases of HFA activity and connectivity. Prior studies have observed single units in rodents and non-human primates with elevated firing rates for positive, negative, and salience RPEs in dMPFC^{10,11,38,54} and dopaminergic midbrain regions^{27,29-33,55}. However, these studies do not typically distinguish these opponent coding schemes for increased activity to positive versus negative RPEs from inverted coding using decreases in activity. This is important to avoid confounding interpretations of signed RPE value⁴⁸. Inverted coding strategies may be more prominent in HFA than single or multiunit data because HFA aggregates both spiking and subthreshold input activity across local populations, making it sensitive to small fluctuations in firing rates and membrane potentials^{52,53}. Furthermore, spiking and HFA are also modulated by low frequency oscillations^{12,56}, which may have different information coding and transmission properties. Future studies examining simultaneously recorded single units, HFA, and LFPs are needed to understand how diverse representations of valence-specific and salience RPEs in individual neurons give rise to asymmetric and inverted coding at the population level.”

Third, regarding recording resolution, high frequency activity (HFA) measured using these rare human intracranial EEG (iEEG) recordings provides greater spatial (~3mm full-width-at-half-max) and temporal (millisecond) resolution compared to widely used non-invasive methods. Although the nominal spatial resolution of standard human fMRI data acquisition parameters is similar, smoothing introduced in post processing typically results in lower resolution data. Moreover, iEEG is not just a lower spatial resolution version of single unit activity. Rather, it provides unique mesoscale information by aggregating both spiking and synaptic activity of local neuronal populations (Leszczynski et al., 2020 Sci. Adv.), which may explain why inverted coding using HFA decreases were more prominent in our data than prior single unit studies. Notably, Rich & Wallis (2017) reported greater decoding accuracy using HFA than single unit spiking activity, which reinforces the additional information available in the HFA signal. Lastly, an important feature of our analysis is using random slopes in a mixed effects model to characterize individual channel responses, thereby maintaining maximal spatial information. Typical statistical approaches in both fMRI and iEEG estimate average responses across multiple channels or voxels and thus focus on region- rather than population-level effects. Thus, our study combines the population-level measurement properties of HFA with statistical techniques that maintain mesoscale resolution to

provide a complement to findings using other recording modalities with both finer and coarser spatial resolution.

R2.2: Next, the dMPFC as defined by the authors (a huge part of cortex encompassing many medial aspects of PFC and beyond as it looks like) are of course no homogeneous structure. Essentially, the reader is led to believe that there is theoretical considerations of RPE representations within a cortical extend that is more diverse than most even rudimentary imaging methods. The results are underwhelming. The authors talk about spatial interleaving of different RPE types but what the reviewer sees is non specificity. How can I be sure that the same type of response is not found in other prefrontal regions making the entire exercise futile in terms of specificity? Do the authors actually believe these are unique representations?

We acknowledge that our dMPFC ROI includes multiple subregions, though as noted in the Results, coverage in our dataset is primarily in mid-cingulate with some nearby supplementary motor areas and relatively few anterior cingulate sites (Fig. 1a). Although we do not have sufficient power to test for differences between fine grained parcellations of dMPFC, we did test whether our results differed across X, Y, and Z axes for HFA categories and the anterior-posterior axis for HFA connectivity (see response to R3.8 below). We found none but note that this does not imply homogeneity of RPE coding in regions outside dMPFC and INS, which remains to be tested. As the reviewer suggested, we cannot argue specificity to dMPFC and INS and agree with the reviewer that examining RPE coding in other prefrontal regions is of great relevance and hope our study sparks future efforts in this direction.

- Results (lines 142-145): “We collected behavioral data from 10 patients while recording from implanted SEEG and ECoG electrodes in dMPFC (primarily mid-cingulate cortex with some nearby supplementary motor complex and anterior cingulate sites) and INS (Fig. 1a; see Table 3 for patient demographics, electrode coverage, and behavior).”
- Results (lines 324-327): “**The proportion of categories did not significantly change along any of the three spatial dimensions (x: $p \geq 0.09$, y: $p \geq 0.26$, z: $p \geq 0.2$), suggesting that they** were spatially interleaved. **This indicates** mixed coding of RPE features across the cortical surface of both regions (Fig. 3d).”
- Results (lines 410-414): “Finally, channel pairs involved in RPE communication between INS and dMPFC were also spatially interleaved **and category proportions did not change depending on the subregion of dMPFC involved (anterior vs posterior; Sup. Analysis 1)**, which agrees with the aforementioned mixed coding scheme for RPEs in neuronal populations (Fig 4c).”
- Supplement (lines 963-975):

- **“Supplemental Analysis 1. Analysis of differences in RPE categories between anterior and posterior dMPFC for connectivity estimates**
- The dMPFC ROI used in the main analyses includes multiple subregions. As an initial assessment of differences across subregions, we tested whether patterns of RPE categories were different between anterior and posterior divisions of dMPFC. Using linear mixed-effects models to predict peak lags and comparing them through likelihood ratio tests, we confirm no significant differences between anterior (y MNI coordinate < 12) and posterior (y MNI coordinate \geq 12) dMPFC connections for either negative RPE coefficients ($\chi^2(6) = 4.62, p = 0.59$) or positive RPE coefficients ($\chi^2(6) = 1.92, p = 0.93$). Similarly, there are no significant interactions between subregion and channel pair category (positive RPE coefficients: $\chi^2(24) = 14.2, p = 0.94$; negative RPE coefficients: $\chi^2(24) = 11.45, p = 0.99$).“

Regarding the impact of our findings, we believe it is striking that the proportion of populations representing different RPE variables changes little across the extended dMPFC region and even between insula and dMPFC. This confirms that RPE representations are distributed across the salience network and aligns with research that RPEs are globally relevant signals that are broadly distributed in many brain regions (e.g., Vickery et al., 2011 Neuron). However, we do observe relatively greater proportions of unsigned and positive RPEs in dMPFC and INS, and these RPE categories also showed different directional connectivity patterns from INS to dMPFC that were not observed for signed and negative RPEs. Furthermore, at the region level, there was a bias in both INS and MPFC towards positive RPE estimates, which parallels the effect of positive RPEs on behavioral RT adjustments and demonstrates the functional relevance of our neural findings. Thus, we agree with the reviewer that these results reflect the broad availability of task-relevant information across distributed cortical regions but also emphasize that the different proportions of RPE representations and their distinct directional connectivity patterns reflect some degree of selectivity for the most behaviorally relevant variables according to our behavioral model.

- Discussion (lines 531-538): “Our results also indicate that population activity within and connectivity between dMPFC and INS have stronger representations for positive than negative RPEs in our task. Modeling RT adaptation showed a slowing effect specifically following positive RPEs that occurred above behavioral adjustments explained by the previous RT and outcome. A recent behavioral study showed effort can enhance learning from positive RPEs and suppress learning from negative RPEs⁶⁶, suggesting this neural bias towards positive RPE coding in HFA may reflect the behavioral relevance of those learning signals, particularly in the hard condition.”

R2.3: If the authors actually wanted to test the distributional dopamine value code hypothesis they would test its actual unique prediction: It should be possible to decode the full value distribution. Right now the manuscript is confusingly motivated and at best can be interpreted as a translation of findings from the animal literature into the human domain. To that end the reviewer would however be more interested in actually seeing distributions over the entire prefrontal cortex as could be obtained with different techniques. There is a mismatch between the motivation of the study and the actual delivery.

Following the comments of this and other reviewers, we have made it clear across the manuscript that our aim is not to test the dRL theory, but rather to assess asymmetric RPE responsiveness as a key coding strategy in the salience network. Consequently, we now only refer to dRL in the discussion (see also response to reviewers R1.1 and R4.1, including quoted changes to the manuscript).

Our experiment was designed to disentangle the valence and salience of RPEs, allowing the identification of asymmetric RPE responses. However, the design was not optimized to test other predictions of dRL, such as the ability to decode reward distributions from neuronal activity. For example, we clamped block difficulty ($74.4 \pm 6.9\%$ for hard and $19.5 \pm 2.6\%$ for easy blocks) to manipulate expected value. A design optimized for decoding reward distributions should instead induce expectations for a wider range of reward values to provide conclusive tests of dRL predictions. Furthermore, it is not clear how the quantiles (or expectiles) of the reward distribution at which different neurons converge should be calculated in the case of neuronal populations that exhibit inverse coding schemes (e.g., increasing HFA responses to negative RPEs) as found in our data. We look forward to theoretical developments in this regard, but for these reasons, we refrained from decoding RPE distributions from our dataset.

Finally, while our study has one goal of bridging the gap between animal and human research on RPE coding, it also makes neurophysiological and theoretical contributions. First, the insula is a key region for reward processing that is not typically studied in the rodent or primate literature. Our directional connectivity results suggest a leading role of this area in reward and surprise processing. Second, the existence of inverse coding schemes, as we report in the study, is a conceptual development relevant to both human and animal neurophysiology. Third, our results highlight the flexibility of asymmetric RPE models to disentangle signed, unsigned, and valence-specific RPE responses. Importantly, we believe that adopting similar models that can identify and

adjudicate between these different neural RPE coding schemes may reconcile conflicting findings in the literature and clarify debates in the field. Therefore, as noted by the reviewer, our study connects animal and human findings, but also provides a novel perspective on RPE coding in the salience network. Nevertheless, as noted above, we have reoriented the language in the introduction and the discussion away from comparisons with dRLL to make these contributions more clear.

- **Discussion (lines 538-546):** “This finding fits with evidence from non-human primate single unit studies^{10,17,34} and some human iEEG results⁶⁷. However, a variety of conflicting evidence from other prior studies argues dMPFC and INS show a bias for processing negative valence^{22,41,43,68}. In particular, Gueguen et al. report a bias for negative RPEs in HFA responses in anterior INS⁴¹. **This discrepancy could be due to methodological differences such as region- rather than population-level measures and models that did not control for salience. For example,** many of the individual and pairs of channels that responded to negative RPEs in our analyses were revealed to code for salience once we accounted for their response to positive RPEs.”

Reviewer #3 (Remarks to the Author):

Review Nat Comm 30012023

This paper tested whether and how neural activity in human dorsal medial frontal cortex and anterior insular region encodes reward prediction errors (RPE), and how functional connectivity between the two regions contributes. The study reports analyses of high gamma power (HFA) from sEEG recordings in 10 patients trained in one of 2 versions of a Target time interval timing task. All but one patient had electrode contacts in both brain regions. The task consisted in block of easy or hard trials which were matched for difficulty levels across subjects.

In order to test relationships between HFA and RPE the authors modelled behaviour using a simple logistic regression model to estimate the probability of winning function. Using such estimation of expected reward outcome, they computed various versions of RPE that would refer to different theoretical propositions about signals observed in those brain regions and about how REP is putatively computed by the brain.

In summary the authors conclude that asymmetric positive and negative RPE coding better explained feedback-related activity in dMPFC and INS, suggesting a bias towards positive reward outcomes in these areas, HFA suggest a bias towards positive reward outcomes in these areas, and that non-linear heterogeneous representations of reward information are the dominant coding scheme. Finally, communication for pRPE and uRPE flowed predominantly from INS to dMPFC. This is a well-done study with precious data obtained in human subjects with dual recordings in 2 regions known to be relevant

for adaptive decision making. However, the current manuscript does not clearly demonstrate how novel the results are in face of the existing electrophysiological literature in monkeys and humans which is quite extensive. The most novel results are the one on directional coding. Some major improvement in the data description, including behaviour, might reinforce the points of the authors. See below comments that might help improve the manuscript.

We thank the reviewer for their appreciation of our efforts, and for their insightful comments on ways to improve the manuscript, especially with regard to the description of behavioral data.

R3.1) The first major comment is about the core assumption that, in this task, subjects do fluctuate their own representations of expected value based on performance and that their brain estimates RPE based on that representation. Although the authors discuss the fact that their task employs an implicit win-stay lose-shift strategy, the behavioural analysis does not show that subjects adapt.

There is no behavioural data showing that subjects adjust their performance within blocks to the current difficulty level (i.e. not that they have varying performance but that they react to changes in performance). Maybe analysing the fluctuations of errors in timing could do the trick or at least provide hints on how subjects conceived the task. It would be essential to then make the hypothesis about the use of some RL algorithm by the brain in the task and to test the various models of RPE computations.

In agreement with the reviewer's suggestion, we have analyzed fluctuations in behavior. We estimated the adjustment in response time (RT) of a trial compared to the previous trial (i.e., RT difference between consecutive trials) and estimated how it was influenced by several factors.

- Results:

- Lines 170-176: "We investigated the impact that the outcome of the previous trial had on the performance of the current trial. We used linear mixed modeling to predict adjustments in RT relative to the target based on direct RT and outcome feedback, as well as RPEs (Table 1). First, we established that previous trial RTs predicted the adjustment in the current trial ($\chi^2(1) = 2587.269, p < 0.001$). Second, we observed a main effect of previous trial outcome (win, neutral, loss; $\chi^2(2) = 12.24, p = 0.002$) and an interaction between previous RT and previous outcome ($\chi^2(2) = 43.961, p < 0.001$)."
- See Table 1 (lines 178-183)

- Lines 185-231: “Having established relationships between the two sources of feedback and RT adjustments, we next investigated whether RPEs had an impact on behavior. We compared a null model (with only previous RT and feedback predictors, but no RPEs) against an RPE value model (with a signed RPE predictor), an RPE salience model (with an unsigned RPE magnitude predictor), and an asymmetric RPE model (with separate positive and negative RPE magnitude predictors). We found that the asymmetric RPE model predicted RT adjustments better than the null model ($\chi^2(2) = 13.441, p = 0.001$) and the RPE value model ($\chi^2(2) = 10.746, p = 0.001$). Compared with the RPE salience model, the asymmetric RPE model fit was not significantly different ($\chi^2(1) = 2.556, p = 0.109$), although it performed slightly better according to Akaike Information Criterion (AIC) (Table 1). To corroborate these results and further adjudicate between RPE salience and asymmetric RPE models, we replicated our analyses using RT data from a larger sample of healthy participants ($n = 32$) performing the same task during a previously published EEG experiment (see Methods)⁴⁹. Using the enhanced statistical power in this prior dataset, we found that the asymmetric RPE model predicted RT adjustments better than the RPE salience model ($\chi^2(1) = 58.888, p < 0.001$), providing evidence for valence-dependent effects of RPE on behavior.
- Coefficients in the asymmetric RPE model of the current iEEG dataset indicated that RT inversely predicted the adjustment in the following trial ($\beta = -0.75, p < 0.001$; Fig. 1e; see Sup. Table 1 for full report of parameters). Thus, if a participant was early, the following RT tended to be longer, whereas if a participant was late, the following RT tended to be shorter, bringing RTs closer to target. An interaction between previous RT and outcome showed this effect was larger following losses ($\beta = -0.27, p = 0.001$), suggesting a win-stay/lose-switch strategy. Lastly, a slowing of RTs was observed after positive ($\beta = 0.02, p < 0.001$) but not negative ($\beta = 0.01, p = 0.22$) RPEs, supporting a positive bias in the impact of surprising outcomes on behavior. Each of these effects was replicated in the larger sample of healthy participants (Fig. 1f and Sup. Table 2).”

○
 ○ “Figure 1: *iEEG* recording sites, task design, and behavioral modeling. (a) Reconstruction of *iEEG* recording sites in dMPFC (top) and INS (bottom)

across all participants plotted on a standardized group brain after mirroring all channels to the right hemisphere for dMPFC and left hemisphere for INS. (b) Participants pressed a button to estimate the time when lights finished moving around a circle. The gray target zone cue displayed error tolerance around the 1 s target interval. Audiovisual feedback is indicated by the tolerance cue turning green for wins and red for losses. A black tick mark displayed RT feedback. For 4 patients, blue neutral feedback was given with no RT marker on 12% of randomly selected trials. (c) Tolerance and outcome data for an example participant. Larger markers show block level accuracy; smaller markers show binary single trial outcomes. Model fit using logistic regression provides single trial estimates of win probability, which is converted to expected value. (d) Predictions for RL model predictors. Error bars indicate standard deviation between participants. (e) Effect of previous RT (left) and previous RPE magnitude (right) on current trial RT change. (f) Convergent results are shown for a larger behavioral dataset from a previous study⁴⁹. Slopes depict regression coefficients. Shaded areas depict 95% confidence intervals. pRPE = positive reward prediction error, nRPE = negative reward prediction error.”

In addition to rational adjustments to being early/late, participants tended to adjust more after receiving negative feedback than positive or neutral feedback. This result provides evidence for a win-stay lose-switch strategy that modulates adaptation to the continuous RT feedback. Furthermore, slowing following positive RPEs constitutes a clear behavioral correlate of difficulty-dependent RPE processing. Importantly, this effect was observed **after** controlling for the main effects of the previous RT and feedback.

Finally, we point the reviewer to additional data and results on subjective ratings of win probabilities from our previous EEG study (Hoy et al., 2021 *Communications Biology*) that demonstrate participants' trial-by-trial expectations (and thus RPEs) are modulated by difficulty manipulations. This prior manuscript reports a behavioral dataset in which participants rated their subjective probability of winning using a slider bar after responding but before feedback. These results are described in Supplemental Note 2 and Supplemental Figure 10. To briefly summarize, we found strong overall correspondence between subjective ratings and model-based outcome predictions that reflected differences within and between easy and hard blocks. Specifically, participants reported higher ratings of win probability before wins than losses overall and within only hard and only easy blocks, indicating reward expectations were modulated by task difficulty both at the condition level (i.e., easy versus hard blocks) and within condition (i.e., trial-by-trial fluctuations). Rating data did reveal subjective biases relative to the behavioral model, in that participants overestimated their likelihood of winning in hard conditions and underestimated it in easy conditions. However, incorporating the most extreme examples of subjective biases from the rating data did not change the observed brain-behavior relationships, suggesting that the results in the current manuscript from single-trial regressions predicting neural power using RPE features should not be affected by these subjective biases. Therefore, prior evidence suggests participants are sensitive to both the difficulty manipulation and their own behavior, which combine to form reward expectations in our task. We thank the reviewer for motivating these additional analyses.

R3.2) If I understand well the authors did not use an RL model but a logistic regression incorporating error tolerance to estimate expected values across a session. This means it is not exactly a single trial estimation but a global estimation adjusted for trial difficulty of each trial; it does not consider local variations of performance due to fatigue, lapses of attention, and does not account for local rate of outcomes. It also assumes a learning rate equal to 1. I understand this is a simplified RPE but it might, or not, have important local differences with RL types of RPE. Some local variations of HFA at outcomes can be observed in figure 2. If this is correct then could the authors discuss whether local variations in performance could impact their analyses of HFA?

We agree with the reviewer that local variations in behavior should affect RPE computations and their neural representations, and we hope that the new results from modeling behavioral RT adjustments, as well as their replication in a larger behavioral dataset from a prior EEG study, provide convincing evidence of some of these effects. Unfortunately, we do not have measures of fatigue or lapses of attention and thus cannot measure how these variables might influence our findings. We would note that our task was designed to be completed in around 20 minutes to minimize disruptions in the Epilepsy Monitoring Unit. We carefully monitored patient behavior, and the task provided rest breaks in between each of the four blocks to avoid fatigue, which were self-paced. In the second version of the task, a minimum break duration of greater than 15 seconds was enforced. In contrast, behavioral tasks designed to capture fluctuations in attention often use longer tasks to ensure robust measurements of those phenomena. Thus, future experiments using designs optimized to detect these effects are needed to address interactions between lapses in attention and RPE coding. We added a sentence on the potential effects of attention lapses or fatigue on our results.

- Results (lines 148-149): “Easy and hard trials were presented in separate blocks with self-paced breaks in between to minimize fatigue.”
- Discussion (lines 546-552): “Additionally, RPE representations are likely influenced by features of our timing task, including interactions between effort and reward driven by different control demands across easy/hard conditions⁶⁶, the absence of learning effects precluding use of traditional temporal difference RL algorithms to estimate value, differences between positive versus negative punishment (i.e., delivering aversive stimuli versus omitting positive rewards), or potential effects of attention and fatigue.”

Regarding the local rate of outcomes, unpublished control analyses in our prior EEG study found that modeling expected value at the session-level using logistic regression explained the EEG data better than defining expected value as accuracy computed using a rolling average of the last 3 or 5 outcomes. Lastly, as mentioned in the Methods, Results, and Discussion, our task has minimal learning during the trials analyzed, which justifies assuming a learning rate equal to 1.

Finally, as described in our response to R4.2 below, the single trial HFA plotted in Figure 2a is sorted by condition, outcome, and RT, meaning variations in these patterns do not reflect local, trial-by-trial fluctuations. We have added text to the figure caption to clarify this point:

- Fig. 2 caption (lines 255-257): “*Figure 2: Positive and negative RPEs are encoded in a separate, valence-specific manner. (a) Top: Single-trial HFA power at an example channel in the INS is plotted time-locked to feedback (markers at*

feedback indicate condition). Trials are plotted after sorting by condition, outcome, and RT. ”

R3.3) It is a bit hard to get the specifics of the task design maybe because information is spread in the paper. Since there were 2 versions of a task for different patients and with different specifics (cue, feedback, timings) it would have been great to really show the 2 versions and take a bit of writing to detail every task element. The task procedure is hard to follow, e.g. it is quite difficult to understand how the motion cue is presented. For instance, in v2, was the target moving continuously or only lighting separated dots as seems to be shown on figure 1? Was it the same for v1? The description of the bullseye target is also very limited. In addition, the task design presented in Figure 1 is the one describing version 2 of the task which was not performed by the majority of patients. The description of a trial goes like this:” trials began with presentation of a visual motion cue at a constant speed to arrive at a target at the one second temporal interval. Participants estimated the interval via button press“. The verbal instruction was probably not to estimate the interval of 1 second but to estimate when the cue would reach the arrival zone, which might not be exactly the same (i.e. reaching the border of the zone vs. the centre).

We thank the reviewer for pointing out this potential ambiguity, and we have added details to the Methods section to provide more details on both versions of the task and a new Supplemental Figure 1 to show the linear version of the task in a manner similar to the circular version presented in Figure 1b.

- **Methods (lines 681-689):** “In the first version of the task (n = 6), the motion cue was a blue dot moving continuously upwards in a straight line towards a bullseye target (Sup. Fig. 1), and in a second version (n = 4), the motion cue was individual lights flickering on then off again in a counter-clockwise order starting and ending at the bottom of a ring of dots on which a gray target zone was centered (Fig. 1b). Participants were instructed that “Your goal is to respond at the exact moment when the ball hits the middle of the target.” or “...when [the light] completes the circle.” for the first and second versions, respectively. The size of the bullseye in the first version and the width of a gray target zone in the second version indicated the tolerance for successful responses.”
- **Supplemental Figure 1 (lines 917-921):**

- c
- **Supplemental Figure 1: Task version No 1. Participants pressed a button to estimate the time when a ball traveling upwards hit the center of the bullseye target. The size of the bullseye cue displayed error tolerance around the 1 s target interval. Audiovisual feedback is indicated by the text "Win!" in green or "Lose!" in red, along with a tick mark of the same color marking the RT.**

R3.4) The author tested (1223) different patterns of response to positive and negative RPE seeking a validation of models of RPE found in the literature. They report that their data revealed a bias in both regions for unsigned prediction error. Indeed, the overall distribution of significant glmm coefficients (which appears to be most relevant feature) shows a clear extension on the axis of unsigned prediction errors (figure 3b). This comforts the fact that signed RPE cases are the fewest. Regarding the additional bias for pRPE, it is likely that losses/wins in easy and hard have different sources and categorical meanings. The bias for pRPE contrasts somewhat with the literature although that same literature has shown that positive or negative biases are due to tasks statistics and task rules. The bias might in fact reveal that subject do not perceive/use losses as much as they integrate positive outcomes.

We thank the reviewer for this observation, which we now address with newly added behavioral modeling results (see response to R3.1). Our results reveal a mixture of adaptation strategies. First, participants adjust their RTs most after losses, indicating that they do perceive/use losses. This is an important observation to confirm the patients are using rational task strategies (i.e., win-stay, lose-switch) and are engaged in task performance on a trial-by-trial basis. Subjective rating data from healthy controls reported in Hoy et al. (2021) and described above in the response to R3.1 also confirm that participants have dynamic reward expectations that track both task difficulty and current trial performance estimates. Second, there is an additional RT slowing effect following positive RPEs specifically, demonstrating that participants also integrate unexpected positive feedback in a unique manner. This finding parallels the bias for pRPE coding in our neural HFA and connectivity results and provides evidence for the behavioral relevance of this enhanced neural representation. Collectively, these different types of behavioral adaptation confirm the behavioral relevance of the diverse RPE signals identified in our neural data.

- Results: see quoted new text, figures, and table from response to R3.1
- Discussion (lines 440-442): “This positive bias parallels the modulation of RT adaptation by positive but not negative RPEs, which underscores the behavioral relevance of neural RPE coding in these areas.”
- Discussion (lines 531-538): “Our results also indicate that population activity within and connectivity between dMPFC and INS have stronger representations for positive than negative RPEs in our task. Modeling RT adaptation showed a slowing effect specifically following positive RPEs that occurred above behavioral adjustments explained by the previous RT and outcome. A recent behavioral study showed effort can enhance learning from positive RPEs and suppress learning from negative RPEs⁶⁶, suggesting this neural bias towards positive RPE coding in HFA may reflect the behavioral relevance of those learning signals, particularly in the hard condition.”
- Methods (lines 711-757): see “Behavioral modeling” section for details on new linear mixed modeling of behavioral adjustments

R3.5) A general bias in coefficients is clear as well as, in the figure, with a shift towards positive values, i.e. larger increased gamma activity with increased RPE. However, the authors report a high proportion of recording sites with decreasing HFA with increasing positive prediction error. It would be interesting to see a few examples of such decreasing cases. It also raises other questions regarding the activity preceding the outcome response, or other activity potentially reflecting expected value. Did the authors find similar proportion of increasing decreasing HFA with increasing expected values? Actually, for some reasons, analyses of expected value-related HFA are presented only for INS-MPFC coordination not for each region individually.

We agree with the reviewer that these inverted coding schemes are interesting and important to account for in neural analyses and interpretations. To further illustrate the common occurrence of these diverse response profiles, we have added a new Supplemental Figure 2 showing the time courses of random effects coefficients for each electrode for the three regressors of the asymmetric model (expected value, pRPE, nRPE). These plots exemplify both increasing and decreasing HFA coding for all three predictors, indicating these opponent coding schemes may be an intrinsic property of population activity. Note that, although fixed-effect coefficients for expected value at the region level (depicted in Fig 2c) are not significant, some individual electrodes significantly encoded expected value with either increases or decreases of HFA. We now refer to expected value coding in the results section.

- Results (lines 357-359): “Time courses of significant expected value, positive RPE, and negative RPE coefficients for individual channels are plotted in Sup. Fig. 2, demonstrating similar bidirectional coding of expected value in both regions.”

- Supplement (lines 952-956):

- **“Supplemental Figure 2: Time course of individual channel random-effects coefficients for each of the three regressors in the asymmetric RPE model of**

HFA. Colors indicate channel category. Significant timepoints are highlighted in bold.”

R3.6) A related comment about HFA: It is said that they are known to correlate with local multi-unit activity; This is true to the extent that sEEG represents a compound signal of large numbers of neurons especially when compared to MUA recorded with single contact microelectrodes. The compounds signal supposedly mixes local populations of single units that do express various coding schemes. In monkeys, Matsumoto et al. 2007 and Quilodran et al 2008 for instance, have shown that single unit activity in the medial frontal cortex reflect most often separately positive and negative outcomes. (Note: the authors referred to Monosov 2017 for RPE coding and anatomical heterogeneity in cingulate recording locations, but I am not sure that paper actually reported precisely that). Many other studies have shown separate cortical encoding of positive and negative outcomes and the modulation by expectations. Quilodran and co-authors also showed HFA reflecting the phenomena. However only a small proportion of units in these studies did not differentiated between positive and negative outcomes. This might explain why the current studies find that asymmetric coding explains better the data and at the same time that unsigned RPE wins the game – if HFA reflect a compound signal. Maybe the authors could discuss their result in light of these past studies and regarding the nature of HFA vs SUA. Also, some studies showed that non-selective responses to outcomes had lower latencies than the discriminant ones, do the authors find similar effects?

First, we thank the reviewer for their emphasis on the topic of how neural coding of valence and salience is implemented at the single unit and population level, which we feel has important implications for basic neuroscience methodology and interpretation and will impact efforts to identify potential biomarkers for clinical disorders such as addiction, depression, and anxiety. As noted by the reviewer, HFA is a compound signal that mixes multiple forms of local population activity. One major component of HFA is spiking activity (i.e., single and multiunit firing), as reported by Quilodran et al. (2008, *Neuron*), Manning et al., (2009, *J. Neuro.*), Ray & Maunsell (2011 *PLoS Bio.*), and others. A second major contribution to HFA is local synaptic inputs (Leszczynski et al., 2020 *Sci. Adv.*). Therefore, the information content in HFA can be greater than and, in some cases, dissociated from spiking activity due to sensitivity to small changes in membrane potentials and/or synchronous activity (Rich & Wallis 2017).

This aggregation of multiple local neurophysiological phenomena into HFA amplifies signals that may not be significant in single or multiunit activity, which may explain the increased prevalence of inverted coding using decreases in activity for neuronal representations in HFA. We therefore suggest that understanding and disseminating the coding properties of HFA is important to capture its potential for bridging species and

scales in neuroscience. Accordingly, we have added language calling for future studies to examine the relationships between individual neurons and population-level coding.

- Discussion (lines 472-487): “Our analyses also revealed that reward and salience information were represented with both increases and decreases of HFA activity and connectivity. Prior studies have observed single units in rodents and non-human primates with elevated firing rates for positive, negative, and salience RPEs in dMPFC^{10,11,38,54} and dopaminergic midbrain regions^{27,29-33,55}. However, these studies do not typically distinguish these opponent coding schemes for increased activity to positive versus negative RPEs from inverted coding using decreases in activity. This is important to avoid confounding interpretations of signed RPE value⁴⁸. Inverted coding strategies may be more prominent in HFA than single or multiunit data because HFA aggregates both spiking and subthreshold input activity across local populations, making it sensitive to small fluctuations in firing rates and membrane potentials^{52,53}. Furthermore, spiking and HFA are also modulated by low frequency oscillations^{12,56}, which may have different information coding and transmission properties. Future studies examining simultaneously recorded single units, HFA, and LFPs are needed to understand how diverse representations of valence-specific and salience RPEs in individual neurons give rise to asymmetric and inverted coding at the population level.”

We also agree that diverse combinations of single units using regular/inverted coding to represent different signed, valence-specific, or unsigned RPEs could explain our results. For example, one potential bias would be a greater proportion of channels with regular coding using increases in activity than inverted coding using decreases in activity. However, our data provide evidence that all categories of signed, valence-specific, and unsigned RPEs, as well as expected value (Sup. Fig. 2), are represented using inverted coding. Another possibility is that equal numbers of valence-specific positive and negative RPEs exist, and as the reviewer suggests, the combination of these populations explains the overrepresentation of unsigned RPEs in our data. However, given the prevalence of inverted coding, this organization should also give rise to strong signed RPE representations, which are not evident in our data. Furthermore, the bias towards positive RPEs we observed suggests an imbalance between positive and negative RPE representations in the local populations measured in dMPFC and INS. Lastly, previous reports of single units in MPFC coding for unsigned RPEs (e.g., Hayden et al., 2011, J. Neuro.) supports the argument that the prevalence of unsigned RPE effects in our HFA data is not solely due to aggregation of valence-specific RPE coding at the single unit level.

Overall, we agree with the reviewer that considering both the proportions of single units representing different RPE variables and the regular/inverted coding schemes used to do so will be important for understanding neural computations in these networks, but simultaneous single unit and HFA recordings are needed to address this question. We have therefore added language to the manuscript calling for such experiments (as quoted above).

Second, we thank the reviewer for pointing out the ambiguous language used to describe the findings from Monosov (2017). We did not intend to claim this paper, which recorded all units from the same region, showed anatomical heterogeneity, only that it showed heterogeneity in RPE coding within anterior cingulate cortex. We have amended the sentence to clarify this point:

- Discussion (line 559-564): “In contrast, single units recorded in non-human primates from anterior cingulate cortex, which is anterior to dMPFC, show reduced salience coding and mostly responded to positive and negative RPEs³⁴. This difference in the relative strength of signed and unsigned RPE coding is potentially because anterior cingulate cortex is a distinct subregion linked to limbic circuits involved in learning, comparing, and choosing values than action control^{26,70-73}.”

Finally, regarding the reviewer’s comment about faster latencies for non-selective responses, we agree that there is some evidence from dopaminergic midbrain neurons for an initial salience response in the first 200 ms followed by more selective RPE responses (reviewed in Schultz 2016 Nat. Rev. Neuro and Watabe-Uchida, Eshel, & Uchida 2017 Ann. Rev. Neuro.). Following the reviewer’s suggestion, we used mixed effects models and likelihood ratio tests to evaluate possible differences in peak latencies between uRPE and other electrode categories. We find no significant differences between categories for both positive-RPE coefficients ($\chi^2(7) = 5.6, p = .59$) and negative-RPE coefficients ($\chi^2(7) = 2.27, p = .94$).

R3.7) In figure 4C it would be also relevant to know how many patients contribute to each MPFC and INS sub-region. In addition, most recordings in INS and in the medial or posterior part of the INS, apparently not reaching Anterior Insula, which however has been the main analysed region in past studies regarding outcome and RPE analyses. The authors might want to discuss that.

As suggested by the reviewer, we have added a new Supplemental Figure 3 showing the connectivity modulation estimates per subject. We agree that the particular subregions of the MPFC and INS may influence the proportion of valence-specific and salience responses observed in population signals. We discuss this possibility and the

relationship between our results and prior studies with different coverage (e.g., more anterior INS and MPFC) in the Discussion:

- See **Supplemental Figure 3** (lines 957-961) for individual connectivity results.
- **Discussion** (lines 554-575): “Another potential factor influencing the proportion of positive, negative, and salience responses is where our specific recording sites are located relative to functional gradients within dMPFC and INS. For example, the strong representations of salience in our results **may be due to** the majority of our recording sites falling in mid-cingulate and insular cortices overlapping with the salience network, which is associated with control and performance monitoring^{9,57,69,70}. In contrast, single units recorded in non-human primates **from anterior cingulate cortex, which is anterior to dMPFC**, show reduced salience coding and mostly responded to positive and negative RPEs³⁴. This difference in the relative strength of signed and unsigned RPE coding is potentially because anterior cingulate cortex is a distinct subregion linked to limbic circuits involved in learning, comparing, and choosing values than action control^{26,70-73}. Similarly, our results showed some negative RPE coding in the INS that aligns with previous studies reporting a bias towards negative RPEs in the anterior portion of this region^{4,41,68,74,75}. However, **this contrast with the bias towards positive RPE coding found in our INS data may be explained by differences in** spatial sampling, which was determined solely based on clinical needs of the patient **and covered primarily mid- and posterior INS in our dataset**. Interestingly, this potential shift in sensitivity from negative to positive bias across the anterior-posterior axis fits with observations from rodent research of a hedonic “hot spot” in the INS where stimulation induces “liking”, which is found posterior to a hedonic “cold spot” in more anterior INS^{76,77}. Overall, our converging results from both individual channels and between-region connectivity indicate that dMPFC and INS are predominantly modulated by positive RPEs and RPE salience.”

R3.8) Figure 4c related data led the authors to suggest that there might be bidirectional communication. This is very likely indeed, and the data for pRPE and uRPE, reveal quite a number of cases with negative lags. The analyses do not discriminate between subregions but the data (recording sites) clearly clusters at least in MPFC between aMCC regions and pMCC or PCC regions. Could the authors check whether subregions in MPFC lead to different ratio of positive versus negative lags with INS?

Using linear mixed effects models to predict peak lags and comparing them through likelihood ratio tests, we confirm no significant differences between anterior (y MNI coordinate < 12) and posterior (y MNI coordinate ≥ 12) dMPFC connections for either

negative-RPE coefficients ($\chi^2(6) = 4.62, p = .59$) or positive-RPE coefficients ($\chi^2(6) = 1.92, p = .93$). Similarly, there are no significant interactions between subregion and channel pair category (positive-RPE coefficients: $\chi^2(24) = 14.2, p = .94$, negative-RPE coefficients: $\chi^2(24) = 11.45, p = .99$). We report this in Supplemental Analysis 1. We also tested whether X, Y, or Z dimensions predicted RPE category responses of individual channels and found null results (see response to R2.2).

Minor comments.

R3.9: 1) (I312-320) Isn't it expected (statistically) that the proportions of INS/MPFC correlations coefficients for each RPE category were similar to that found in HFA analyses?

We agree with the reviewer and want to add that these similar proportions are a nice sanity check of the coherence of our results.

R3.10 2) Maybe figure 4c would be more straightforward if axes were inverted such that time lags were facing the anatomical representations (i.e. positive time lags would be at the levels of INS->MPFC figures).

We thank the reviewer for their suggestion. Although we agree that the proposed orientation would better match the anatomical representations, we decided to keep the current version of the figure because it is more coherent with other plots in the manuscript in which time-related variables were depicted in the x axis.

Reviewer #4 (Remarks to the Author):

Title: asymmetric coding of reward prediction errors in human insula and dorsomedial prefrontal cortex

Summary: Hoy et al. examine asymmetric scaling of RPEs in epilepsy patients, specifically focusing on high gamma activity in the dorsomedial prefrontal cortex and insula while patients performed a psychometrically regulated time estimation task. The authors find that unsigned and positive RPEs are preferentially encoded, over signed and negative RPEs, and that such activity occurs slightly earlier in insula than dmPFC. This is an interesting study, but is impugned by the lack of controls.

We thank the reviewer for their interest in our study and address their concerns regarding control analyses below.

Additionally, there are some issues with interpretation of the results that should be corrected.

Major critiques:

R4.1: - It is disingenuous to frame the findings of this study as relevant for distributional RL without performing several controls and additional analyses. As is, this manuscript is about asymmetric RPE scaling, not dRL. It is unsurprising that the results don't cohere with dRL given the task design and measurements. If the authors read the Supplementary Information for the Dabney study, they will notice numerous controls that were performed to ensure that they were observing distributional RL, and not some other facet of asymmetric reward prediction scaling. Moreover, the current study doesn't meet the basic criteria for establishing the presence of distributional RL: 1) diverse RPE scaling, 2) diverse reversal points, and 3) correlation between 1 and 2. Furthermore, distributional RPE coding at the level of neuronal populations (~0.5 million neurons near an sEEG) is much less tractable, harder to understand. It begs the question, why did the authors mention dRL at all, especially given the observed results (relatively little negative PRE scaling, compared to unsigned RPE scaling)? I would recommend only mentioning dRL in the discussion, if at all, with extensive discussion of the caveats involved with the current study: dRL is not defined at the level of neuronal populations, the authors didn't show distributional encoding, the authors didn't estimate reversal points, or show any neural or behavioral evidence for asymmetric learning.

As mentioned above in responses to R1.1 and R2.3, we agree with this reviewer and others that our study does not directly address the criteria for establishing dRL. We have taken this reviewer's advice and reframed the study to only mention dRL in the Discussion, which now includes a discussion of the caveats involved in linking our current findings to dRL theory:

- Please see modified text quoted from the Introduction, Results, and Discussion sections in R1.1

Note that our new behavioral analyses do address the reviewer's last concern by showing behavioral evidence for asymmetric learning. Specifically, we show larger RT adjustments following losses and behavioral slowing selectively following positive RPEs, which are both valence-specific and occur over and above RT adjustments related to the previous response being early/late.

- See quotes from Results, Methods, and Discussion sections in our response to R3.1

R4.2 - In Figure 2, the largest high gamma values correlate with the density of task structure changes. One can even see increases in high gamma earlier in the largest blocks. It therefore seems important to control for the potential surprise induced by changes in the structure of the task, especially, given previous evidence that ACC is sensitive to changes in task structure. The authors should also show the time courses of the TD model variables. How did the authors account for an initial learning period that is ubiquitous in TD learning models.

We acknowledge the reviewer's concern about potential surprise driven by changes in task structure. We would like to clarify that Figure 2 plots trials after sorting by condition and RT, not by order of presentation. Importantly, easy and hard trials were presented in separate blocks (2 of each), meaning large transitions in reward probability were rare in the task. Participants were also familiarized with the reward probabilities in easy and hard blocks from their experience during the staircase initialization procedure conducted separately for easy and hard conditions at the beginning of the task, which reduces the uncertainty following transitions between conditions. Furthermore, outcomes varied on a trial-by-trial basis, so the rare trial types plotted in the middle rows of Figure 2 are interspersed throughout the task. It is unlikely that changes in task structure can explain our results, including surprise responses. Nevertheless, the reviewer's comment prompted us to clarify these details of task structure in the caption of Figure 2 and emphasize them in the manuscript.

- **Results (lines 145-151):** "These patients performed an interval timing task that dissociates valenced RPE value and non-valenced RPE magnitude by using task difficulty manipulations to modulate reward expectations (Fig. 1b). Easy and hard trials were presented in separate blocks with self-paced breaks in between to minimize fatigue. Error tolerance was adjusted after each trial by two staircase algorithms to clamp accuracy at $74.4 \pm 6.9\%$ and $19.5 \pm 2.6\%$ (mean \pm SD) in easy and hard blocks, respectively."
- **Fig. 2 caption (lines 255-257):** "Figure 2: Positive and negative RPEs are encoded in a separate, valence-specific manner. (a) Top: Single-trial HFA power at an example channel in the INS is plotted time-locked to feedback (markers at feedback indicate condition). Trials are plotted after sorting by condition, outcome, and RT. "
- **Methods (lines 672-674):** "The interval timing task was written in PsychoPy⁸⁵ (v1.85.3) and consisted of four blocks (two easy and two hard) of 75 trials each (see Fig 1A for task schematic), with an initial instruction cue before each block started indicating the difficulty level."
- **Methods (lines 700-703):** "Note that our design minimizes surprise related to task transitions due to the blockwise nature of the difficulty manipulation, presentation

of an explicit cue for difficulty level (“Easy”/“Hard”) before each block started, and participants’ learning of reward probabilities during training.”

We have also emphasized in the manuscript that our model is not based on TD learning but on a simplified model based on logistic regression which used all the data at once without incorporating dynamics. The use of this model is justified by the fact that there is little dynamic learning going on in the task for two reasons. First, three phases of training are used to introduce the task (fully visible trials) and then to titrate difficulty via staircase procedure separately for easy and hard training trials. Second, the expected outcome of a trial is easily inferred from the size of the tolerance at the beginning of a block. Instead, as revealed by our behavioral analyses (see response to R3.1), during the block there are small adjustments of expectations around the mean value in response to RPEs, which are captured by our model. Furthermore, the practice trials (35 in total) allowed participants to fine tune tolerance-outcome associations before the main task. Nevertheless, we now explicitly acknowledge that estimating RPEs using logistic regression instead of temporal difference RL algorithms may influence our results:

- **Methods (lines 729-732):** “Note that our task minimizes learning by providing an explicit tolerance cue (gray target zone) on each trial after the initial expectations are learned during easy and hard training blocks. Consequently, values were estimated using a logistic regression model instead of traditional temporal difference RL algorithms.”
- **Discussion (lines 546-552):** “Additionally, RPE representations are likely influenced by features of our timing task, including interactions between effort and reward driven by different control demands across easy/hard conditions⁶⁶, the absence of learning effects precluding use of traditional temporal difference RL algorithms to estimate value, differences between positive versus negative punishment (i.e., delivering aversive stimuli versus omitting positive rewards), or potential effects of attention and fatigue.”

R4.3 - The interpretation on line 219 indicates that the asymmetric model performs best, but it appears to perform similarly to the unsigned RPE model. Furthermore, the unsigned RPE model is the most prevalent. More justification for the authors interpretation is required.

We performed likelihood ratio tests to compare the asymmetric model with both the RPE value model and the RPE salience model, revealing significant differences for all time points, after FDR correction. However, note that, due to the large number of data points, our analyses are overpowered to detect even small differences, which renders p-

values uninformative. For this reason, instead of p-values, we rely on AIC values to assess model performance in the manuscript. A clarification has been added to the methods:

- Methods (line 817-822): “We compared three different RL models using AIC as a performance metric. Note that, due to the large amount of data points, likelihood ratio tests resulted in significant differences between the models at all time points, even after FDR correction, thereby rendering p-values uninformative. We therefore relied on AIC values for model comparisons.”

Furthermore, the reviewer points to an apparent discrepancy between the best model (asymmetric RPE) and the most prevalent category of single channel responsiveness (unsigned RPE). We note that, besides the prevalence of uRPE channels, pRPE channels were far more abundant than nRPE channels, which is in agreement with the bias towards positive responses in fixed-effects coefficients of the asymmetric model.

- See quoted text from Results, Discussion, and Methods sections in responses to R3.1 and R3.4

R4.4 - Activity in dmPFC has repeatedly been associated with mental effort and cognitive control. An essential control analysis should therefore account for variance introduced by trial difficulty. It is not clear whether this analysis is possible, given the task design of the current study, as trial difficulty highly correlated with outcome surprise. To what extent does the task design promote expression of these particular flavors of RPEs. This concern is amplified by the observed lack of expected value encoding. One might expect that a brain region responsible for the computations implicated in this study would also represent expected value variables. That these brain areas are not representing expected value raises the concern that they may be representing some other type of surprise rather than task related RPE. Maybe shifting the model fits by one trial to account for subsequent trial effects, or examining responses to errors, could speak to how much of the high gamma activity is induced by cognitive control demands?

The reviewer raises an interesting point regarding a possible confound between cognitive control demands and RPE responses in our design. First, we note that trial difficulty is an essential feature of our paradigm: Our difficulty manipulation controls participants' expectations of winning or losing, which was designed specifically to decorrelate difficulty, valence, and outcome surprise (i.e., easy losses, hard wins, and neutral outcomes in both conditions are all equally likely). We then incorporate difficulty in our model and results via the expected value predictor because our study was designed to assess diversity of RPE responses rather than isolate the effects of task difficulty. Nevertheless, to further address the reviewer's concern, we tested whether

adding difficulty (easy vs. hard trials) as a factor in behavioral models explained additional variance and found no significant differences with the best-performing asymmetric model ($\chi^2(1) = 1.19, p = 0.27$). Regarding the encoding of expected value signals, it is true that we did not observe significant fixed effects of expected value at the region level (Fig. 2c, Fig. 4a), but we do observe individual channels with significant expected value coding. We have added a new Supplemental Figure 2 to display these data and language in the Results to clarify this point.

- Results (lines 357-359): “Time courses of significant expected value, positive RPE, and negative RPE coefficients for individual channels are plotted in Sup. Fig. 2, demonstrating similar bidirectional coding of expected value in both regions.”

While we do observe channels encoding expected value at feedback, such signals are typically examined in value-based decision making paradigms during cue presentation epochs, when the necessary information for this computation becomes available. In our timing task, the cue indicating the target zone size (which determines expected value) on each trial was presented simultaneously with the onset of the timing interval, meaning expected value analyses in the traditional cue epoch would be confounded by interval timing and motor preparation activity. We constrain our analyses to the feedback epoch, when RPE signals are dominant.

Finally, we have added new language to the Discussion to address control demands as an additional factor when interpreting our results:

- Discussion (lines 546-552): “Additionally, RPE representations are likely influenced by features of our timing task, including interactions between effort and reward driven by different control demands across easy/hard conditions⁶⁶, the absence of learning effects precluding use of traditional temporal difference RL algorithms to estimate value, differences between positive versus negative punishment (i.e., delivering aversive stimuli versus omitting positive rewards), or potential effects of attention and fatigue.”

Minor critiques:

R4.5 - It seems like the sentence on line 50 should be supported with citations.

We have added several references to support this claim.

- Introduction (lines 55-58): “However, conflicting reports linking dMPFC and INS activity to a diverse range of signed and unsigned RPE signals have fueled long-

standing theoretical debates about their role in reward learning and cognitive control¹⁰⁻¹⁵“

R4.6 - The introduction could be more concise. Removing the irrelevant dRL mentions and focusing more on asymmetric scaling may help with this.

We agree with the reviewer, and we have removed any mention of dRL in the Introduction.

R4.7 - The models fit the unsigned RPE data better, yet the effects were smallest for unsigned RPEs. The authors should discuss this discrepancy.

We apologize for the confusion. To clarify, the unsigned RPE model fit the data better than the signed RPE model, but was not better than the asymmetric RPE model, largely because the asymmetric RPE model captures both signed and unsigned RPEs. Note that our model comparison metric, the Akaike Information Criterion (AIC), shows better model performance as lower values, which is why the asymmetric RPE model is lowest in Fig. 2b and the unsigned RPE model is in between the winning model and signed RPE. At the individual channel level, the unsigned RPEs were the largest (coefficients in Fig. 3b and Fig. 4b) and most common (percentages in Fig. 3c) effects. We hope this clarifies the reviewer’s concerns and confirms the lack of discrepancy in these results. We have added language to the manuscript describing how the asymmetric RPE model captures unsigned RPE responses to prevent misunderstanding from readers:

- Introduction (lines 135-137): “These results resolve competing theories of dMPFC function by demonstrating that asymmetric coding enables both valence-specific and unsigned RPE salience signals to coexist within overlapping dMPFC and INS circuits, ...”
- Discussion (lines 492-495): “Importantly, we leveraged the flexibility allowed by asymmetric coding to categorize combinations of positive and negative coefficients corresponding to the key variables underlying central theories of dMPFC function, including RPE value and salience.”
- Discussion (lines 592-594): “In conclusion, our results demonstrate that incorporating asymmetric coding principles can capture positive, negative, and salience RPE coding in human dMPFC and INS.”

R4.8 - The anatomical claims in the discussion (line 474) are unsupported by data.

We thank the reviewer for catching this ambiguous phrasing. We did not mean to imply our data shows differences across INS sub-regions. In fact, we show no differences in RPE effects across subregions of dMPFC or INS (see response to R2.2 and R3.8). We

have rephrased the sentence to clarify that it is our spatial sampling, not the RPE effects, which are primarily concentrated in mid- and posterior INS.

- Discussion (lines 566-569): “However, this contrast with the bias towards positive RPE coding found in our INS data may be explained by differences in spatial sampling, which was determined solely based on clinical needs of the patient and covered primarily mid- and posterior INS in our dataset.”

R4.9 - Line 662: did the authors confirm that their data matched the model assumptions after log-normalizing, with a Lilliefors’s test for example?

We visually inspected the residuals through qq-plots and histograms to evaluate conformity of the data with model assumptions. Note that, because we have so many data points, a Lilliefors’ test would detect the smallest deviations from normality, which would not be informative.

The inspection revealed that HFA residuals were slightly skewed toward positive values. This is expected given that the original power data is nonlinear. Thus, while the log normalization to baseline mitigated this issue, some nonlinearity remained. This slight skewness does not have major consequences for model inferences, because: 1) we have used a robust estimation method in which data points substantially departing from model assumptions are given less weight; 2) we have a large amount of data points; 3) the regularizing properties of mixed-effects models prevent against false positives; 4) the t -statistic is typically robust to deviations from normality, specially for large datasets.

Nevertheless, for comparison, we have done a sensitivity analysis in which the data was transformed to “rankit” values (see Supplemental Analysis 2 and Supplemental Figures 4, 5, and 6 for details) to ensure conformity with the gaussian residuals assumption. This procedure led to converging results. We used the same approach for connectivity data, whose residuals were heavy-tailed.

R4.10 - Line 667: ‘feedback-lock’ should be ‘feedback-locked’

Thank you for catching the typo. It has been corrected.

R4.11 - General: Both linear mixed-effects models are superficially described. It’s unclear if both patient and channel variables were nested relative to the signals, or there was a hierarchical nesting of channel and patient variables. This issue could be clarified by specifying the models in either matrix or Wilkinson notation.

Indeed, there was a hierarchical nesting of channels within subjects for random effects. To improve the clarity of the details of our linear mixed-effects modeling, we have added the model structures in Wilkinson notation in Tables 1 and 2.

Table 1. Linear mixed effects modeling of behavioral data. Model structure is given in Wilkinson notation. Each model was compared with a “null” model excluding the parameter of interest. Degrees of freedom (Df), chi-squared statistics (χ^2) and p-values (p) are reported for likelihood ratio tests. Akaike Information Criterion (AIC) is also shown for each model. pRPE = positive reward prediction error, nRPE = negative reward prediction error, sRPE = signed reward prediction error, uRPE = unsigned reward prediction error, sub = subject.

Model	Name	Wilkinson notation	Null	Df	χ^2	p	AIC
b0	Intercept only	RT change ~ 1 + (1 sub)					-1217.4
b1	Previous RT	RT change ~ previous RT + (1 + previous RT sub)	b0	3	2587.27	< 0.001	-3798.6
b2	Outcome	RT change ~ previous RT + outcome + (1 + previous RT sub)	b1	2	12.24	0.002	-3806.9
b3	Interaction	RT change ~ previous RT*outcome + (1 + previous RT sub)	b2	2	43.96	< 0.00	-3846.8
b4	RPE value	RT change ~ previous RT*outcome + sRPE + (1 + previous RT sub)	b3	1	2.69	0.1	-3847.5
b5	RPE salience	RT change ~ previous RT*outcome + uRPE + (1 + previous RT sub)	b3	1	10.88	0.001	-3855.7
b6	Asymmetric RPE	RT change ~ previous RT*outcome + pRPE + nRPE + (1 + previous RT sub)	b3	2	13.441	0.001	-3856.3

Table 2. Structure of linear mixed effects model included in the HFA analyzes in Wilkinson notation (EV = expected value, HFA = high frequency activity, pRPE = positive reward prediction error, nRPE = negative reward prediction error, sRPE = signed reward prediction error, uRPE = unsigned reward prediction error, sub = subject, chan = channel).

Model name	Wilkinson notation
Signed RPE	HFA ~ sRPE + EV + (1 + sRPE + EV sub) + (1 + sRPE + EV sub:chan)

Unsigned RPE	$HFA - uRPE + EV + (1 + uRPE + EV \text{sub}) + (1 + uRPE + EV \text{sub:chan})$
Asymmetric RPE	$HFA - nRPE + pRPE + EV + (1 + nRPE + pRPE + EV \text{sub}) + (1 + nRPE + pRPE + EV \text{sub:chan})$

REVIEWERS' COMMENTS

Reviewer #1 (Remarks to the Author):

The authors have successfully addressed my prior comments, all of which were relatively minor. I think this paper makes an important contribution to the literature.

Reviewer #2 (Remarks to the Author):

The authors did a very thorough job of revising the manuscript and have tried to be responsive to the reviewers critiques.

Unfortunately, my main criticism has not been adequately addressed and stands. The manuscript still makes it seem like there is a lack of consensus among RPE theories in PFC which is just not correct. It is correct that further experimentation is clearly warranted and I am happy the authors are doing that but the manuscript is overselling its position in the field and its impact on an issue that is constructed for the purpose of the paper.

Lastly the authors have essentially changed the narrative of the paper dramatically and if i follow the new reasoning its impact would be better in a more specialized journal.

Reviewer #3 (Remarks to the Author):

The authors basically answered with good care most of my comments and did a great job in clarifying the paper which really needed major revisions. I am still unsure about the novelty of the outcome especially because it is uncertain whether the neurophysiological signals analysed here are adequate to really resolve the major point, i.e. the neural encoding of prediction errors.

In any case, the study provides interesting data and analyses that are now fully comprehensible.

I would just like to pass on to the authors a few comments because, for what it's worth, I disagree with some of their statements:

- The supposed higher resolution of sEEG is not what they suggest. The 'resolution' referred to is the neural volume represented by the recorded HFA, but there is no good resolution in the context of functional mapping, because sEEG cannot provide mapping information, and for the very important reason that recording sites should be decided for medical reasons for the benefit of patients, and not for scientific reasons and goal to produce a scientifically interesting high-resolution mapping of a brain regions. Mapping with good resolution is way better with surface ECoG grids. So, in that sense the spatial resolution of sEEG is not as good as fMRI for instance because one cannot have a good single subject functional mapping. In addition, I don't want to be too picky about anatomical specificity, because these are data difficult to acquire, but one has to acknowledge that (as for fMRI mapping) the mixing of subjects makes things quite complicated. Unless there is local non-linear co-registration of brain regions (based on sulci for instance) between subjects it is really difficult to equate coordinates even in a common space. The error is made by most brain imaging studies (i.e. 95% of the publications), and if one compares single subject locations of functional subregions across subjects, even in a common standard space, one observes enormous spatial variability due to morphological inter-individual differences. Hence, statistics along x, y, z of sEEG responses is really not convincing, unless this is done within subjects.

- The comparison of Single Unit decoding with High Frequency Activity decoding in Rich and Wallis 2017 cannot be compared with HFA with sEEG. The former was acquired with microelectrodes and from LFP, a way more local signal than the one obtained from sEEG probes used in humans. In addition, one should compare decoding of HFA with decoding of a population of simultaneous single units decoded as a whole which was not the case in that study.

Reviewer #4 (Remarks to the Author):

The authors have clarified my misunderstandings and have made thorough edits in response to my critiques.

Hoy, Quiroga-Martinez, et al. Asymmetric RPE Nature Communications final revision:
Response to reviewers

Reviewer comments in black. Author responses in blue. Modified manuscript text in red.

Reviewer #1 (Remarks to the Author):

The authors have successfully addressed my prior comments, all of which were relatively minor. I think this paper makes an important contribution to the literature.

We are glad to have addressed the reviewer's comments and thank them for their kind words.

Reviewer #2 (Remarks to the Author):

The authors did a very thorough job of revising the manuscript and have tried to be responsive to the reviewers critiques.

Unfortunately, my main criticism has not been adequately addressed and stands. The manuscript still makes it seem like there is a lack of consensus among RPE theories in PFC which is just not correct. It is correct that further experimentation is clearly warranted and I am happy the authors are doing that but the manuscript is overselling its position in the field and its impact on an issue that is constructed for the purpose of the paper.

Lastly the authors have essentially changed the narrative of the paper dramatically and if i follow the new reasoning its impact would be better in a more specialized journal.

We thank the reviewer for considering our manuscript once again and providing their feedback. We agree that further experimentation is needed. However, it seems that we have some disagreement regarding the consensus of RPE theories in PFC and the impact of our work. In our previous response letter, we argued that:

1. The human literature is mixed and inconclusive due to methodological challenges and modeling confounds.
2. Animal research often uses tasks and models that do not dissociate effects of valence and salience, as well as whether information is coded using increases or decreases in activity.
3. High frequency activity provides better spatial and temporal resolution than many other human recording and imaging methods and unique information not captured by single unit recordings.

When considered collectively and applied in the context of our results, these points reveal that population level neurophysiological responses measured at the level of high frequency activity have additional information coding formats and capacities than single units. In particular, (1) the increased range offered by coding with decreases in HFA not available to single units (due to the floor in firing rate) and (2) the separation of RPEs by valence enable simple yet flexible

opportunities to read out information, both for the brain and for potential neurotechnologies such as brain-computer interfaces. Methodologically speaking, it is still common for papers to predict brain activity using fully linear signed RPE regressors, leading to improper interpretations of what may be asymmetric (and/or unsigned) RPE coding and highlighting that many researchers may not share the reviewers theoretical clarity in this field.

While researchers using animal models and particularly single unit recordings may be more familiar with diverse RPE coding (e.g., signed vs. unsigned, asymmetric positive and negative, etc.) within one region, the human RPE coding literature commonly focuses on identifying a single, primary function, representation, or coding scheme for a given brain region or circuit, including signed/linear RPEs. Our work captures RPE coding diversity using mesoscale HFA measurements in humans, thereby overcoming methodological and species differences to bridge these two fields. To the extent that human neuroscience is critical for the ultimate goal of understanding the brain in health and disease, we feel this is an important contribution.

We would also add that the reviewer correctly notes that the nature of RPE coding in dmPFC has been extensively studied in both humans and animals. However, our finding that RPE signals in the insula precede and predict those in the dMPFC adds an important element that should shift the focus of the field away from a singular focus on dMPFC. For example, few laboratories deploy resources to record in the rodent or monkey insula, and our findings suggest insula recordings should be addressed in the field regardless of how well RPE coding in dMPFC is understood. Therefore, our manuscript addresses an important gap in knowledge and makes a valuable contribution to the literature.

Nevertheless, per the reviewer's feedback, we have toned down our claim of lack of consensus in prefrontal RPE theories:

- **Abstract edit 1:** "... conflicting evidence for both signed and unsigned activity has led to **multiple** proposals for the nature of RPE representations in these brain areas."
- **Abstract edit 2:** "These findings support asymmetric coding across distinct but intermingled neural populations as a core principle of RPE processing and **inform theories** of the role of dMPFC and INS in RL and cognitive control."
- **Introduction paragraph 1:** "However, conflicting reports linking dMPFC and INS activity to a diverse range of signed and unsigned RPE signals have fueled **multiple theoretical proposals** about their role in reward learning and cognitive control¹⁰⁻¹⁵."
- **Introduction paragraph 3:** "One barrier to **adjudicating between** these **different** theories is that studies often assume positive and negative RPEs are represented together on a symmetric, linear scale relative to a single mean expected value."
- **Discussion:** "Overall, these findings bridge region-level analyses common in human neuroscience with population-level analyses in animal models and **inform theories** debates regarding neural coding of RPEs in dMPFC and INS."

Reviewer #3 (Remarks to the Author):

The authors basically answered with good care most of my comments and did a great job in clarifying the paper which really needed major revisions. I am still unsure about the novelty of the outcome especially because it is uncertain whether the neurophysiological signals analysed here are adequate to really resolve the major point, i.e. the neural encoding of prediction errors. In any case, the study provides interesting data and analyses that are now fully comprehensible. I would just like to pass on to the authors a few comments because, for what it's worth, I disagree with some of their statements:

- The supposed higher resolution of sEEG is not what they suggest. The 'resolution' referred to is the neural volume represented by the recorded HFA, but there is no good resolution in the context of functional mapping, because sEEG cannot provide mapping information, and for the very important reason that recording sites should be decided for medical reasons for the benefit of patients, and not for scientific reasons and goal to produce a scientifically interesting high-resolution mapping of a brain regions. Mapping with good resolution is way better with surface ECoG grids. So, in that sense the spatial resolution of sEEG is not as good as fMRI for instance because one cannot have a good single subject functional mapping. In addition, I don't want to be too picky about anatomical specificity, because these are data difficult to acquire, but one has to acknowledge that (as for fMRI mapping) the mixing of subjects makes things quite complicated. Unless there is local non-linear co-registration of brain regions (based on sulci for instance) between subjects it is really difficult to equate coordinates even in a common space. The error is made by most brain imaging studies (i.e. 95% of the publications), and if one compares single subject locations of functional subregions across subjects, even in a common standard space, one observes enormous spatial variability due to morphological inter-individual differences. Hence, statistics along x, y, z of sEEG responses is really not convincing, unless this is done within subjects.

We thank the reviewer for their feedback and are pleased that their concerns were properly addressed.

Regarding the resolution of iEEG, we want to clarify a few points. We do not claim any one human method is superior but would like to provide our views on the strengths and weaknesses of these research tools.

1. The combined spatial and temporal (i.e., "**spatiotemporal**") resolution of iEEG is excellent compared to human neuroimaging methods such as EEG, where localizing the source of scalp potentials is difficult, and fMRI, where temporal resolution is limited.
2. We agree that the **electrode coverage** of sEEG is sparse, as it is purely determined by clinical reasons, and as such it is not an optimal method for the functional mapping of each participant's brain. However, note that our aim was not whole-brain mapping of

single participants but to investigate RPE coding and communication in the salience network (INS and dMPFC). In this regard, we believe we succeeded because:

- a. We obtained a good coverage of both regions (106 electrodes in dMPFC and 64 in INS).
 - b. We used mixed-effect models to estimate the association between RPE coding and HFA. These models take into account participant- and electrode-specific correlations and variability, yielding generalizable statistical inferences that characterize entire brain regions.
 - c. Our connectivity estimates were obtained within each subject, demonstrating the ability of sEEG to provide insights into the communication between brain regions with millisecond precision.
 - d. Except in relatively rare cases with high density grids (~3-4 mm spacing), ECoG electrodes typically have larger spacing (10 mm) than the sEEG electrodes used here (3-5 mm). Further they do not provide spatial sampling as in fMRI with grids often covering an 8x8 cm expanse of cortex with 64 electrodes. Furthermore, ECoG implantations are becoming rarer as neurosurgical workflows continue to shift largely to sEEG (Abou-al-Shaar et al., 2018, *J. Clin. Neurosci.*), thus limiting the potential for the type of mapping described by the reviewer. More importantly, much of the brain is fundamentally inaccessible to ECoG electrodes because a majority of the human cortex is located in sulci, and for local signals in the insula, medial temporal region, and orbitofrontal cortex, sEEG provides excellent coverage (Anderson et al., 2021 *Sci. Reports*). It should also be noted that for the purposes of investigating the salience network, ECoG grids are never placed on the insula for monitoring epileptic activity. These factors collectively limit the mapping potential of ECoG relative to sEEG.
3. Regarding anatomical specificity, we localized individual electrodes for each subject separately and made sure that they were within the specified regions of interest. This was done on participant-specific native MRI coordinates employing consensus between an expert neurologist and functional and anatomical landmarks in the literature. We agree with the reviewer that within each region there is substantial variability in functional response profiles, which is an important finding from our work with implications for future studies articulated in the Discussion section. Furthermore, previous animal single unit and human iEEG studies have reported substantial within-region heterogeneity of information coding, indicating this variability may be a consistent property of neural organization across scales. Unfortunately, we cannot assess spatial gradients in individual subjects due to the sparse nature of coverage in standard clinical iEEG implantations, which does not allow for adequate sampling along hypothesized functional gradients (e.g, anterior-posterior insula or dMPFC). Regarding non-linear co-registration and alignment across participants, we agree that differences in brain morphology will be critical factors in differentiating specific subregions, but given the above mentioned limitations of sparse within-subject sampling, our analyses grouped electrodes according to larger dMPFC and INS regions that are less sensitive to variation in individual anatomy. We look forward to future research exploring the anatomical specificity of reward coding with higher spatial detail.

We have added a clarification of these points in the manuscript:

- Discussion: "Further research with denser sampling within these regions may reveal the fine-grained spatial organization of these RPE variables across subregions."

- The comparison of Single Unit decoding with High Frequency Activity decoding in Rich and Wallis 2017 cannot be compared with HFA with sEEG. The former was acquired with microelectrodes and from LFP, a way more local signal than the one obtained from sEEG probes used in humans. In addition, one should compare decoding of HFA with decoding of a population of simultaneous single units decoded as a whole which was not the case in that study.

We thank the reviewer for highlighting this important point. Unfortunately, a direct comparison of HFA recorded with sEEG and SUA is still lacking from the literature. Therefore, to our knowledge, Rich and Wallis (2017, Nat. Comm.) and Leszczynski et al. (2020, Sci. Adv.) are the best existing references regarding the relationship between HFA and SUA. Importantly, our results show coding diversity across sEEG electrodes, suggesting that information about reward is present at the population level as well. We look forward to future research addressing the relationship between sEEG and SUA, which indicate HFA is a local signal that indexes both multi-unit activity and dendritic potentials.

Reviewer #4 (Remarks to the Author):

The authors have clarified my misunderstandings and have made thorough edits in response to my critiques.

We are glad to have clarified the reviewer's concerns.